

# Bridging Gas and Aerosol Properties between Northeast U.S. and Bermuda: Analysis of Eight Transit Flights

Cassidy Soloff[1], Taiwo Ajayi[1], Yonghoon Choi[2,3], Ewan C. Crosbie[2,3], Joshua P. DiGangi[2], Glenn S. Diskin[2], Marta A. Fenn[2,3], Richard A. Ferrare[2], Francesca Gallo[2], Johnathan W. Hair[2], Miguel Ricardo A. Hilario[1], Simon Kirschler[4,5], Richard H. Moore[2], Taylor J. Shingler[2], Michael A. Shook[2], Kenneth L. Thornhill[2], Christiane Voigt[4,5], Edward L. Winstead[2,3], Luke D. Ziemba[2], Armin Sorooshian[1,6]

[1]Department of Hydrology and Atmospheric Sciences, University of Arizona, Tucson, AZ, 85721, USA
[2]NASA Langley Research Center, Hampton, VA, 23681, USA
[3]Analytical Mechanics Associates, Hampton, VA, 23666, USA
[4]Institute of Atmospheric Physics, German Aerospace Center, Germany
[5]Institute of Atmospheric Physics, University Mainz, Germany
[6]Department of Chemical and Environmental Engineering, University of Arizona, Tucson, AZ, 85721, USA

*Correspondence to*: Armin Sorooshian (armin@arizona.edu)





**Abstract.** The western North Atlantic Ocean is strongly influenced by continental outflow, making it an ideal region to study the atmospheric transition from a polluted coastline to the marine environment. Utilizing eight transit flights between NASA Langley Research Center (LaRC) in Hampton, Virginia and the remote island of Bermuda from NASA's Aerosol Cloud meTeorology Interactions oVer the western ATlantic Experiment (ACTIVATE), we examine the evolution of trace gas and

25   aerosol properties off the U.S. East Coast. The first pair of flights flew along the wind trajectory of continental outflow, while the other flights captured a mix of marine and continental air mass sources. For measurements within the boundary layer (BL), there was an offshore decline in particle $N_{<100\,nm}$, $N_{>100\,nm}$, $CH_4$, $CO$, and $CO_2$ concentrations, all leveling off around ~900 km offshore from LaRC. These trends are strongest for the first pair of flights. In the BL, offshore declines in organic mass fraction and increases in sulfate mass fraction coincide with increasing hygroscopicity based on f(RH) measurements. Free troposphere

30   measurements show a decline in $N_{<100\,nm}$ but other measured parameters are more variable when compared to the prominent offshore gradients seen in the BL. Pollution layers exist in the free troposphere, such as smoke plumes, that can potentially entrain into the BL. This work provides detailed case studies with a broad set of high-resolution measurements to further our understanding of the transition between continental and marine environments.



## 1 Introduction

Earth's surface is dominated by marine environments, which interface with continental and anthropogenic air masses. The properties of aerosols in these marine environments have major impacts on public welfare, biogeochemical cycles, weather, and climate, yet there are large uncertainties in particle emissions, formation, evolution, and removal, which complicates modeling of their direct (Stier et al., 2013) and indirect (Pierce and Adams, 2009) effects on climate. Aerosol particles directly attenuate incoming solar radiation (Robinson, 1962) and indirectly through cloud interactions (Twomey, 1977; Albrecht, 1989). Enhanced aerosol concentrations at a fixed cloud liquid water content (LWC) can brighten (Twomey, 1977) and potentially prolong cloud lifetimes (Albrecht, 1989), reducing the fraction of solar radiation that reaches the surface. Combined, the direct and indirect effects contributed to 76% of sea surface temperature anomalies in the North Atlantic from 1860 to 2005 (Booth et al., 2012). Pristine environments with low aerosol number concentrations, like those thought to be present over remote ocean areas, are most responsible for uncertainty in aerosol-cloud radiative forcing (Gryspeerdt et al., 2023). This raises questions as to what is considered pristine and whether some remote marine areas are truly pristine due to enhanced aerosol loadings from long-range transport of pollutants from continents. For instance, Bermuda in the northwest Atlantic can be impacted by sources as far as North America, Africa, and Europe (Smirnov et al., 2002; Smirnov et al., 2000; Keene et al., 2014; Dadashazar et al., 2021). This motivates a look at one of those transport corridors, specifically between the U.S. East Coast and Bermuda to characterize trace gas and aerosol behavior as air transitions from the polluted coastline of a major populated continent to over 1000 km away over the open ocean.

Past studies examining flight data between continental and remote marine environments derived important information on the spatial evolution of aerosol microphysical and optical properties (e.g., Peter and John, 1985; Hansen et al., 1997). The northwest Atlantic has a rich history of atmospheric research with a heavy focus on continental outflow (Sorooshian et al., 2020). It is well documented that the U.S. northeast coast is a significant source of pollution due to urban emissions from many populated cities, industrial and agricultural activity, and biomass burning (Corral et al., 2021). The latter study summarized how seasonally dependent factors (e.g., meteorology, atmospheric circulation, relative amount of emissions from different sources) conspire to modulate aerosol and trace gas composition advected offshore, but that in general when there is offshore flow there is usually a pollution gradient as demonstrated well by carbon monoxide (CO). Noteworthy is a declining trend in CO and related anthropogenic pollutants (e.g., $SO_2$, $NO_x$, sulfate) over the region in recent decades owing to regulatory activities (Corral et al., 2021; Painemal et al., 2021; Keene et al., 2015; Keene et al., 2014; Feng et al., 2020; Jongeward et al., 2016; Hand et al., 2012). One of the earliest studies over the northwest Atlantic from the summer of 1979 made measurements at Wallops Island, Virginia (VA) and found that small (< 0.07 μm diameter) particle concentration decreased with increasing transport time over water (Hoppel et al., 1984). That same year, the Western Atlantic Ocean Experiment (WATOX) began to investigate the sources, evolution, and sinks of continental pollution outflow over the northwest Atlantic (Galloway et al., 1987). One WATOX airborne study between Delaware and Bermuda observed that $SO_2$ decayed by 20% in the boundary layer



(BL) while sulfate particle concentration remained constant (Hastie et al., 1988). Another WATOX study involved flying perpendicular to North American outflow to observe enhanced ozone and lower aerosol scattering in the free troposphere (FT) compared to the boundary layer (Bridgman et al., 1988). They also reported higher extinction and aerosol concentrations in

the BL, except during long-range pollution transport events in the FT (Bridgman et al., 1988). Bermuda was of great importance to these studies because trajectory modeling showed that air masses arriving at the island originated from North America almost 60% of the time (Miller and Harris, 1985). Harriss et al. (1984) used an airborne lidar in transit from Delaware to Bermuda to map out the outflow of continental aerosol, with a key conclusion being that there were up to seven distinct aerosol layers from the sea surface up to 3 km in their case flights. Bermuda was continuously used for future campaigns

including the Global Change Expedition/Coordinated Air-Sea Experiment/Western Atlantic Ocean Experiment (GCE/CASE/WATOX) (Kim et al., 1999), the Bermuda Atlantic Time-series Study (Lomas et al., 2013), the Atmosphere/Ocean Chemistry Experiment (AEROCE) (Arimoto et al., 1999), the Tropospheric Aerosol Radiative Forcing Observational Experiment (TARFOX) (Russell et al., 1999), and the Western Atlantic Climate Study (WACS) (Quinn et al., 2014). Recent studies based on models, reanalysis, and either surface or space-borne remote sensing over the northwest

Atlantic (Aldhaif et al., 2021; Dadashazar et al., 2021; Braun et al., 2021) motivate the need for airborne data to better characterize the spatial and vertical nature of trace gases and aerosols between North America and Bermuda.

The overarching goal of this work is to bridge studies focused on both the U.S. East Coast and Bermuda by using a unique inventory of flight data to complement the aforementioned studies, which featured sparse statistics from airborne platforms and were conducted years to decades ago. This study will examine trace gas and aerosol characteristics as a function of altitude

and offshore distance towards Bermuda based on eight transit flights, with results being broadly relevant to other marine regions where continental outflow can influence trace gas and aerosol properties (e.g., outflow from East Asia, northern and southern Africa, South America). The paper is structured as follows: (i) a description of data collection and analysis methods; (ii) an overview of results including trajectory modeling and examination of flight data as a function of offshore distance and altitude between Virginia and Bermuda; and (iii) conclusions.

## 2 Methods

### 2.1 Flight campaign details

NASA's Aerosol Cloud meTeorology Interactions oVer the western ATlantic Experiment (ACTIVATE) included 162 joint research flights (RF) with two spatially coordinated aircraft across six deployments in winter and summer seasons between 2020 and 2022 (Sorooshian et al., 2023). The region of focus was the northwest Atlantic with the base of operations for most

flights being NASA Langley Research Center (LaRC) in Hampton, Virginia. The low-flying Falcon aircraft (< 3 km) made in situ measurements of trace gases, cloud, aerosol, and meteorological properties above and within the boundary layer during




the following flight legs flown in a stair-stepping manner (Dadashazar et al., 2022b): MinAlt = minimum altitude the Falcon could fly at (~150 m above sea level), BCB = below cloud base, ACB = above cloud base, BCT = below cloud top, ACT = above cloud top, BBL/ABL = slightly below/above boundary layer top in cloud-free conditions. The Falcon sometimes

conducted slanted vertical profiles from the MinAlt level to well above the boundary layer top to as high as ~5 km. The higher-flying King Air aircraft (~9 km) conducted measurements below itself with a lidar, polarimeter, and dropsondes. The focus of this study is eight of the joint aircraft flights, which were one-way transit flights between Hampton, Virginia and Bermuda rather than ACTIVATE's traditional out-and-back flights based out of Hampton (Fig. 1). Three pairs of the transit flights (research flights [RFs] 142-143, 156-157, and 159-160) were on the same day (i.e., to and from Bermuda with refueling at

Bermuda in between), while the other two (RFs 161 and 179) were flown on the ends of conducting a full deployment based in Bermuda in June 2022 (i.e., to Bermuda on 31 May 2022 and back to Hampton on 18 June 2022). The transit flights provide a detailed look at the aerosol and gas properties across a large swath of the northwest Atlantic and how conditions change moving from the continent to a more remote marine area less influenced by both the terrestrial boundary layer and the Gulf Stream situated close to the U.S. East Coast.

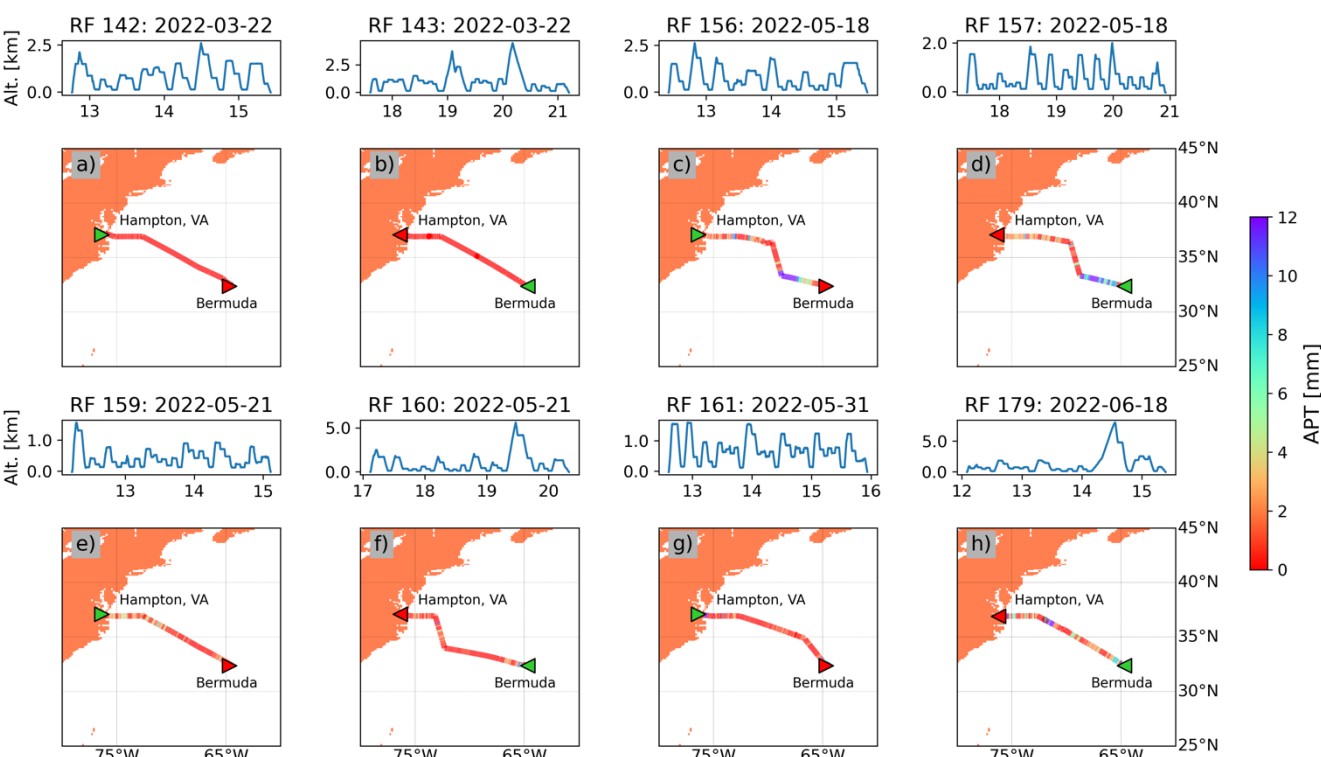

Figure 1: Flight tracks for all eight transit flights along with UTC hour time series of Falcon flight altitude above each map. The flight path is color-coded by the accumulated precipitation along the trajectories (APT) leading to the Falcon altitude along its flight track (APT explained more at end of Sect. 2). The green (red) points mark the start (end) of flights with arrows pointing in the flight path direction.




## 2.2 Instrument data

Here we focus on the instruments and measured variables used for this study (Table 1) but the complete overview of ACTIVATE instruments and data can be found in Sorooshian et al. (2023). Note that some instrument data are unavailable during beginning periods of flights leaving Bermuda due to the inability to keep instruments on during refueling on some days, resulting in the start-up of instruments when flights began and thus delayed stabilization for a subset of instruments.

The King Air High Spectral Resolution Lidar – generation 2 (HSRL-2) (Hair et al., 2008; Burton et al., 2018) obtained vertically resolved aerosol optical properties. Volume aerosol backscatter coefficient (e.g., fraction of light backscattered per sample range and solid angle) at 532 nm is used here to understand the vertical distribution of aerosol particles. The vertical distribution of aerosol type is also used based on the method of Burton et al. (2012). Mixed layer heights are derived using the method of Scarino et al. (2014). Also from the King Air, we use dropsonde data collected with the National Center for Atmospheric Research (NCAR) Airborne Vertical Atmospheric Profiling System (AVAPS). As summarized by Vömel et al. (2023), the dropsondes collected vertically-resolved data below the King Air for relative humidity, static air and dew point temperature, pressure, and wind ($u$, $v$, and $w$ components).

The following in situ measurements are used in this work from the Falcon aircraft. The diode laser hygrometer (DLH) measured water vapor concentration using an open-path, near-infrared absorption spectrometer (Diskin et al., 2002). A Picarro model G2401-m measured CO, $CO_2$, and $CH_4$ concentrations (DiGangi et al., 2021). Particle size distribution data were gathered using a scanning mobility particle sizer (SMPS; Model 3085 DMA, Model 3776 CPC, and Model 3088 Neutralizer; TSI Inc.) for particles with a diameter ($D_P$) between 3-100 nm and a laser aerosol spectrometer (LAS, TSI Inc. Model 3340) (Moore et al., 2021) for particles with $D_P$ between 100-5000 nm. To stitch SMPS and LAS distributions, adjustments were made to the SMPS to allow its native electrical mobility diameter to adapt to the LAS optical diameter; note that the SMPS bin centered at 89.1 nm was omitted to allow for cleaner stitching. Scattering coefficient measurements were made with nephelometry (TSI Inc. Model 3563) and f(RH) was calculated by taking the ratio of scattering between dry (<40%) and humidified (80%) RH conditions using a pair of nephelometers (Ziemba et al., 2013). Absorption coefficient measurements were made with a Radiance Research Particle Soot Absorption Photometer (PSAP). Note that the scattering measurements are for submicron aerosol whereas absorption data are for particles with diameter < 5 μm. A High Resolution Time of Flight Aerosol Mass Spectrometer (HR-ToF-AMS; Aerodyne) (DeCarlo et al., 2008) obtained submicron mass concentrations of non-refractory aerosol species including organics, $SO_4^{2-}$, $NO_3^-$, $NH_4^+$, and $Cl^-$. We also make use of the mass spectral markers m/z 43 and 44, which are representative of mixed hydrocarbons and oxidized hydrocarbons, respectively (DeCarlo et al., 2008). Cloud condensation nuclei (CCN) measurements were made with a CCN spectrometer (Droplet Measurement Technologies [DMT] Inc.) (Moore and Nenes, 2009). The Fast Cloud Droplet Probe (FCDP, SPEC Inc) measured concentrations of supermicron particles and cloud droplets between 3-50 μm (Kirschler et al., 2022; Kirschler et al., 2023).



Aerosol mass and number concentrations are reported at standard temperature and pressure by correcting for atmospheric density.

## 2.3 Calculation details

To account for cloud contamination (i.e., shattered droplets) biasing gas and aerosol data, we omitted data with FCDP liquid water content (LWC) above 0.005 g m$^{-3}$. Although clouds were sporadically present in these flights, this LWC threshold was never exceeded and only flight legs outside of clouds were used in this study. This study is concerned with changes during continental outflow towards Bermuda and thus to conduct analysis and visualization based on distance from LaRC, data were sampled in 15-minute bins. In subsequent figures, median values are displayed along with error bars that represent 25$^{th}$ and 75$^{th}$ percentiles. The same statistics are reported in the text in the following format: $[\text{median}]^{+[75^{th} \%ile - \text{median}]}_{-[\text{median} - 25^{th} \%ile]}$. Percent changes

are calculated with raw values and rounded to the nearest percent while the values reported are rounded based on the uncertainty of the instrument (see Table 1). Aerosol size distribution fits were made using a non-linear least squares fit of a two-mode lognormal distribution with the Python package scipy.optimize.curve_fit (Virtanen et al., 2020).

**Table 1: Overview of the utilized instruments and measured or retrieved variables, categorized based on aircraft platform.**

| Instrument | Measured Variable | Uncertainty | Sample Size Range | Resolution | Reference |
|---|---|---|---|---|---|
| King Air | | | | | |
| High Spectral Resolution Lidar – generation 2 (HSRL-2) | Aerosol backscatter coefficient (355, 532, and 1064 nm) | ~0.2 Mm$^{-1}$sr$^{-1}$ | N/A | 30 m x 6 km | Hair et al. (2008); Burton et al. (2015); Burton et al. (2018) |
| | Aerosol type | N/A | N/A | 135 m x 6 km | Burton et al. (2012) |
| | Mixed layer height | ~100 m | N/A | 15 m x 1 km | Scarino et al. (2014) |
| Vaisala NRD41 dropsonde | Pressure, temperature, dew point temperature, relative humidity, wind (u, v, w components) | P: 0.5 hPa; T: 0.2 °C; RH: 3%; u/v wind: 0.5 m s$^{-1}$; w wind: 1 m s$^{-1}$ | N/A | ~11 m vertically | Vömel et al. (2023) |
| Falcon | | | | | |
| Diode laser hygrometer (DLH) | Water vapor concentrations | 5% | N/A | <0.05 s | Diskin et al. (2002) |
| Picarro model G2401-m | CO, CO$_2$, CH$_4$ concentrations | CO: 5 ppb; CO$_2$: 0.1 ppm; CH$_4$: 1 ppb | N/A | 2.5 s | DiGangi et al. (2021) |
| 2B Tech. Inc. model 205 | O$_3$ concentrations | 6 ppb | N/A | 2 s | Wei et al. (2021) |
| TSI Scanning Mobility Particle Sizer (SMPS); model 3085 DMA, Model 3776 CPC, and model 3088 Neutralizer | Size resolved particle concentrations | 20% | 3 – 100 nm | 45 s | Moore et al. (2017) |



| TSI-3340 Laser Aerosol Spectrometer (LAS) | Size resolved particle concentrations | 20% | 100 – 5000 nm | 1 s | Moore et al. (2021) |
|---|---|---|---|---|---|
| TSI-3563 nephelometer with 80% humidification | Scattering coefficient (450, 550, 700 nm), f(RH) (550 nm) | 20% | < 1 μm | 1 s | Ziemba et al. (2013) |
| Radiance Research Particle Soot Absorption Photometer (PSAP) | Absorption coefficient (470, 532, 660 nm) | 15% | < 5 μm | 1 s | Mason et al. (2018) |
| High Resolution Time of Flight Aerosol 545 Mass Spectrometer (HR-ToF-AMS) | Speciated mass concentrations | <50% | 0.06 – 0.6 μm | 25 s | DeCarlo et al. (2008) |
| DMT Cloud Condensation Nuclei (CCN) spectrometer | CCN concentration at different supersaturations | 10% | < 5 μm | 1 s | Moore and Nenes (2009) |
| SPEC Inc. Fast Cloud Droplet Probe (FCDP) | Size resolved particle/droplet concentrations | 15 – 50% | 3 – 50 μm | 1 s | Kirschler et al. (2022) |
| TSI-3776 Condensation Particle Counter (CPC) | Number concentration | 10% | 0.003 – 5 μm | 1 s | Xiao et al. (2023) |
| TSI-3772 CPC | Number concentration | 10% | 0.01 – 5 μm | 1 s | Xiao et al. (2023) |


## 2.4 Air mass back-trajectory data

To determine sources of air masses sampled by the Falcon, three-day back trajectories were obtained using the NOAA Hybrid
Single-Particle Lagrangian Integrated Trajectory (HYSPLIT) model (Stein et al., 2015; Rolph et al., 2017). Trajectories were
obtained at one-minute intervals using archived NCEP Global Forecast System (GFS) 0.25° data and the model vertical
velocity method. Trajectory endpoints were set to the longitude, latitude, and altitude of the aircraft during different points of
each flight. The output of the model gave the location of the parcel every hour, along with the hourly precipitation rate, which
was used to calculate accumulated precipitation along the trajectory (APT) by summing the precipitation values across each
three-day trajectory (e.g., Dadashazar et al., 2021; Hilario et al., 2021).

**3 Results and discussion**

## 3.1 Atmospheric circulation context

HYSPLIT 3-day back trajectories are displayed in Fig. 2 along the flight paths of the eight transit flights to provide a sense of
influential source regions. The first pair of out and back flights (panels A and B) show pure continental outflow along the
flight path. As a result, these first two flights (RF 142-143 on 22 March 2022) will sometimes be referred to hereafter as
"golden flights" since they represent the most ideal conditions for this study. The second pair of flights (panels C and D)
coincide with continental outflow for most of the flight tracks but with marine influence closer to Bermuda. The third pair of





flights (panels E and F) intersect air masses originating from the southeastern U.S. and over the ocean. RF 161 (panel G) had lower velocity outflow off the continent as compared to the 22 March and 18 May flights, whereas RF 179 (panel H) had mostly marine influence originating from the North Atlantic with less continental influence than most other flights. Thus, the

flights represent varying degrees of continental outflow and marine influence. Figure 1 shows that the 18 March flights (RF 156-157) had the highest potential for wet scavenging effects owing to elevated APT throughout the flight. Using HYSPLIT back trajectories from the golden flight, we estimate a transport time off the coast of the U.S. of ~1.5 days in the BL and ~1 day in the FT.

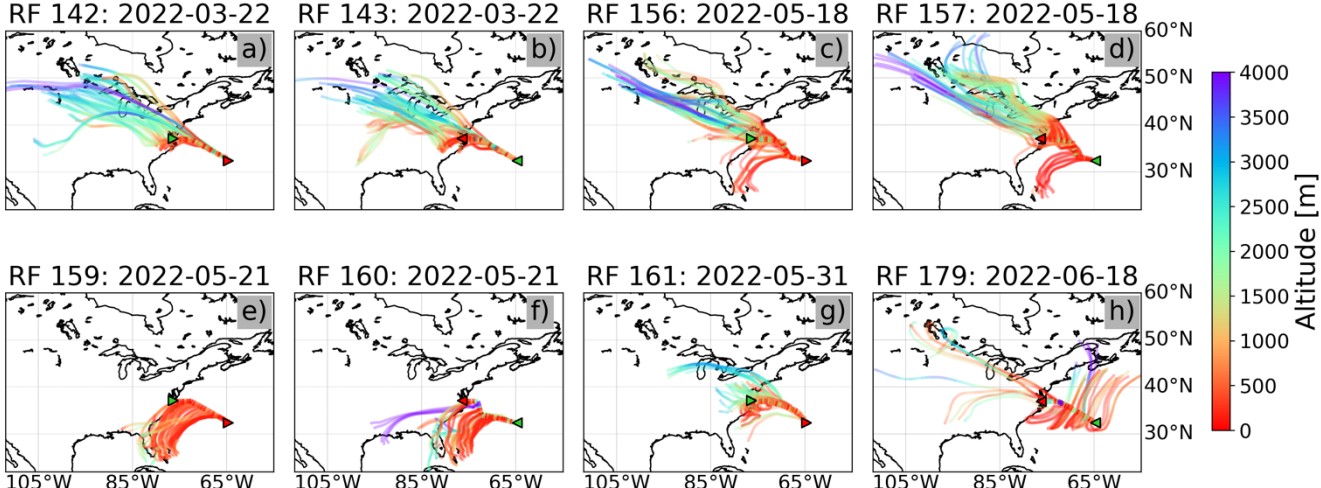

**Figure 2: HYSPLIT three-day back trajectories ending at location of the Falcon aircraft along its flight path. Altitude of air mass shown with color bar.**

**3.2 HSRL-2 perspective of aerosol vertical distribution**

The golden flight day of 22 March 2022 offers a glimpse into the vertical aerosol structure in the ideal case of continental outflow directed along the flight track towards Bermuda. The two sets of flight tracks on this day were essentially identical

but in opposite directions and offset by a few hours (Fig. 1). Figure 3 shows aerosol backscatter vertical profiles for both flights, with enhanced aerosol backscatter in the marine boundary layer. There is a distinct aerosol layer between 1-2 km closer to the Virginia coast, which appears to deepen and disperse more to lower altitudes closer to Bermuda. The layer aloft is more pronounced in backscatter in the morning flight closer to the continent. The Falcon aircraft directly intersects this layer directly in the second flight and intercepts the periphery of the layer in both golden flights. As will be discussed, in situ aerosol and

gas measurements provide evidence of the layer aloft being influenced by smoke. There was also a morning and afternoon pass through the layer aloft near the transition to the boundary layer closer to Bermuda, which is easy to distinguish based on the mixed layer height labels in Fig. 3. For the golden flights, the mixed layer height ranged from $220^{+60}_{-25}$ m within 200 km

from LaRC and, after a steep increase, leveled off around 300 km offshore of LaRC to $990^{+120}_{-120}$ m. The other days showed less

distinct trends with mixed layer height values of $470^{+230}_{-180}$. The other six flights (Fig. S1) exhibit less distinct aerosol layers

except for the 19 May 2022 flight where there was a layer aloft near Bermuda.

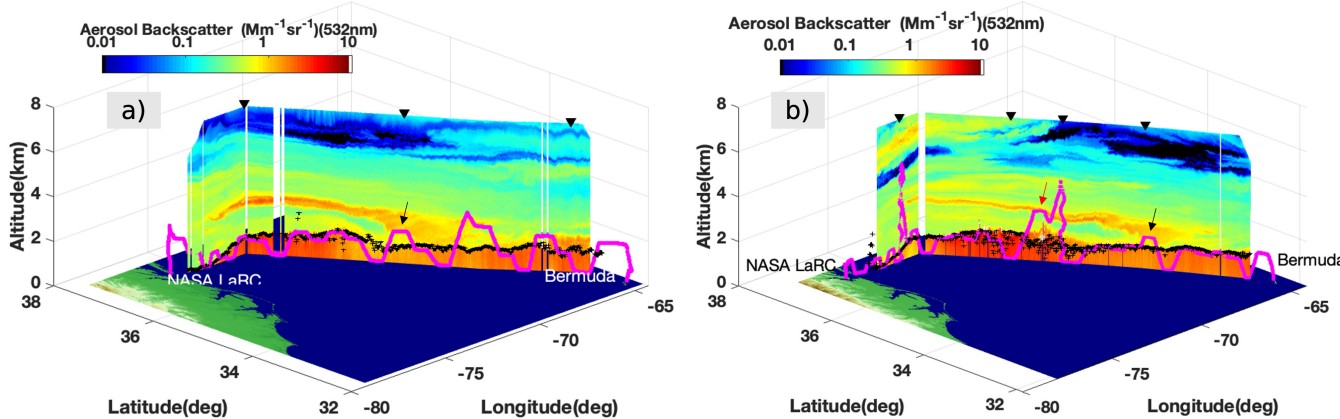

**Figure 3: Aerosol backscatter (532 nm) for the two golden flights on 22 March 2022: RF 142 (a) and RF 143 (b). The magenta lines in each panel represent the HU-25 Falcon flight tracks while the aircraft was spatially coordinated with the King Air collecting these retrievals. Black triangles on top indicate dropsonde launch locations. Black crosses indicate mixed layer height derived from HSRL-2. The red arrow indicates the level leg which directly intersected the smoke layer and the black arrows indicate the level legs where the layer was intercepted near the transition into the boundary layer.**

Aerosol type information inferred from the HSRL-2 retrievals (Burton et al., 2012) suggests that the aerosol layer aloft could

be classified as some combination of "urban/pollution", "dusty mix", and "fresh smoke" (Fig. S2). In the BL, prominent

aerosol types appear to be "marine" closer to Bermuda and some combination of "fresh smoke", "smoke", and "dust mix"

closer to the continent. However, it should be noted that previous ACTIVATE-related work showed that sea salt aerosol (and

thus the "marine" type) can be misidentified as dust owing to it being non-spherical when the relative humidity is depressed

below the deliquescence point of sea salt (~75%) in the boundary layer in the study region (Ferrare et al., 2023); we presume

this is the case during these flights as the RH in the BL was usually below 75% based on Falcon (Fig. S3) and King Air

dropsonde measurements (Fig. S4) and especially on the morning flight when more "dusty mix" was identified. Figure S3

further shows that similarly low RHs were observed on 18 May (RF 156-157), whereas the BL during the last four flights

usually had RH above 75%. This matches well with the aerosol type information from the 18 May flights as "dusty mix" is

even more prominent below 2 km than on 22 March when it is similarly dry in the BL (< 75%), whereas in the BL of the last

four flights, there are different aerosol types in the BL including "fresh smoke", "smoke", "urban/pollution", "marine", and

"polluted marine". Interestingly, on the final transit flight close to Virginia again "dusty mix" is prominent when the RH in



the BL drops well below 75%. These results help provide the spatial and vertical aerosol context and now the Falcon in situ data will be presented with more detailed in situ perspectives of gas and aerosol properties along the flight tracks.

### 3.3 Trace gas results

Gas concentrations of $CH_4$ and CO can be suggestive of continental pollution influence, which we hypothesize should show a decreasing offshore gradient, especially on 22 March 2022. This is generally the case for the flights (Fig. 4), with a clear
decrease in concentration with offshore distance that is most pronounced in the BL. Exceptions include RFs 159-161, consistent with back-trajectories showing less continental outflow influence in favor of more influence from marine sources, which would not be expected to yield a gradient along the offshore flight tracks. For the golden flights, $CH_4$ (CO) concentrations decreased from $2.056^{+0.005}_{-0.006}$ ($0.165^{+0.004}_{-0.003}$) within 200 km from NASA Langley Research Center (LaRC) to $1.989^{+0.001}_{-0.001}$ ($0.131^{+0.003}_{-0.003}$) ppm at >900 km from LaRC. Our investigation is interested in determining if and when the transition
completes from polluted to more remote marine conditions, which we define as being when a continental outflow tracer pollutant levels off. It is noteworthy that the gas concentrations seemed to level off approximately ~900 km offshore for the golden flights. This is coincident with past analysis based on MERRA-2 reanalysis data showing a distinct offshore gradient in tropospheric CO from the U.S. East Coast towards Bermuda, with most of the decline within the first few hundred kilometers offshore (Corral et al., 2021; see Fig. 3). Analogous plots for $CO_2$ and $O_3$ (Fig. S5) show that $CO_2$ concentrations generally
exhibit an offshore decrease in the BL ($433.1^{+1.6}_{-0.7}$ to $423.7^{+0.1}_{-0.2}$ ppm for golden flights measurements <200 km and >900 km from LaRC, respectively) with no clear trend in the FT, whereas $O_3$ concentrations exhibit a less distinct offshore decrease with generally higher values in the FT ($58.8^{+2.3}_{-2.1}$ ppb in FT and $51.5^{+4.2}_{-2.5}$ ppb in BL for golden flights). The latter is consistent with how Corral et al. (2021) showed a different spatial distribution of $O_3$ over the northwest Atlantic relative to CO, with more of a latitudinal gradient as compared to an offshore gradient. Noteworthy also is that surface $O_3$ has been shown to peak
in Bermuda in the spring (March-May) owing to North American pollution transported behind cold fronts (Van Valin and Luria, 1988; Huang et al., 1999; Milne et al., 2000; Li et al., 2002).





**Figure 4: CH₄ (upper half) and CO (lower half) concentrations as a function of offshore distance from LaRC. Blue and orange represent the BL and FT, respectively. Smoke layer median value shown in red for RF 143. Markers represent median values for 15-minute intervals and whiskers are 25th/75th percentiles.**



The enhancement ratio of $CH_4$ to CO provides insight into air mass sources, with lower enhancement ratios suggestive of biomass burning (Wiggins et al., 2021). Enhancement ratios are defined as $\Delta CH_4/\Delta CO$, which can be estimated using the slope of methane versus carbon monoxide concentrations (Fig. 5) (Andreae and Merlet, 2001; Wada et al., 2011). On each flight
day, the enhancement ratios in the BL and FT generally do not indicate evidence of biomass burning, which would require $\Delta CH_4/\Delta CO$ values below 0.3 (Lin et al., 2015). $\Delta CH_4/\Delta CO$ measurements fall between 1–3 and mostly between 1.5-2, typical of urban and ocean emissions (Buchholz et al., 2016; Lin et al., 2015). One exception is from the previously mentioned aerosol layer aloft during the golden flights (RF 142-143) where there was a reduced $\Delta CH_4/\Delta CO$ ratio, representative of biomass burning. The $\Delta CH_4/\Delta CO$ within the smoke layer is 0.45±0.03. Using the highest 70 CO measurements out of the 86 total
measurements made within the layer, corresponding to values with CO concentrations greater than 0.168 ppm (blue line in Fig. 5b), $\Delta CH_4/\Delta CO$ decreases to 0.21±0.06, which is well below the 0.3 biomass burning threshold. The higher CO measurements may represent a more direct sampling of the smoke layer, which could explain the lower $\Delta CH_4/\Delta CO$ ratio. The similar ratios across days suggest somewhat similar types of emissions sources impacting these flights. While there were no significant differences between the BL and FT on a given day, there are still subtle differences in how flight day cluster data
points line up in Fig. 5 owing to different air mass histories.

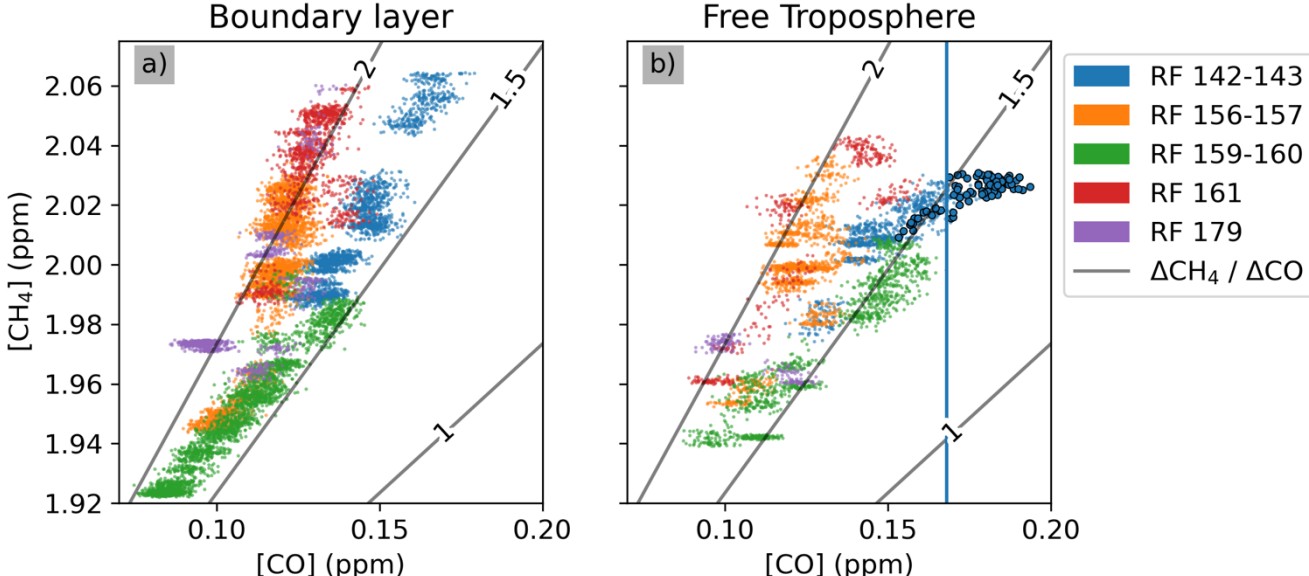

**Figure 5: Ratio of CH₄ to CO concentration for all eight transit flights for the (a) boundary layer and (b) free troposphere, with different colors assigned to each distinct flight day (some days have two flights). Enhancement ratio of CH₄ to CO (ΔCH₄/ΔCO) estimates (1, 1.5, 2) are displayed for comparison using the intercept from a linear fit of the golden flight BL measurements (1.78**
**ppm). The blue points (b) for RF 143 (22 March 2022) outlined in black are measurements made within a smoke layer in the FT (see Sect. 3.8). The solid blue line in panel b represents a CO concentration of 0.168 ppm.**



### 3.4 Particle size distribution behavior

#### 3.4.1 Number concentrations

For particles with $D_p < 100$ nm, there was a clear decrease in particle number concentration ($N_{<100\ nm}$) with offshore distance

both in the BL and FT (Fig. 6). The golden flight (RF 142-143) BL values ranged from $4250^{+260}_{-410}$ cm$^{-3}$ within 200 km of LaRC

to $1150^{+150}_{-70}$ cm$^{-3}$ for distances >900 km from LaRC (a 73% reduction). $N_{<100\ nm}$ seemed to level off around ~900 km offshore

in the BL, consistent with CO and CH$_4$ for the flights less affected by marine air mass influence (Fig. 4). In the FT, $N_{<100\ nm}$

transitioned from $3810^{+700}_{-380}$ cm$^{-3}$ to $1140^{+200}_{-210}$ cm$^{-3}$ between <200 km and >900 km from LaRC (a 70% reduction).

Number concentrations of particles with $D_p > 100$ nm ($N_{>100\ nm}$) decreased with offshore distance within the BL for the golden

flights and RF 179, but was less pronounced for other flights (Fig. 7); similar to Fig. 6, $N_{>100\ nm}$ seemed to level off around

~900 km offshore for the former three flights when a clear gradient was evident in the BL for the golden flights. The golden

flight BL values ranged from $1800^{+60}_{-540}$ cm$^{-3}$ within 200 km of VA to $270^{+20}_{-18}$ cm$^{-3}$ for distances >900 km from VA (an 85%

reduction). In the FT, there is no clear trend across flights.

Supermicron particle number concentrations, as measured by the FCDP in cloud-free conditions, increased with offshore

distance in the BL for the golden flights and RFs 156-157 (Fig. 8). This is in contrast to the sub-100 nm particle number

concentrations and agrees with HSRL-2 data showing enhanced aerosol backscatter and "marine"

aerosol types farther offshore closer to Bermuda. No clear trends are found in the FT likely owing to reduced sea salt. These

results are consistent with an increasing sea salt loading in the BL as we move offshore and a lack of noticeable sea salt

detrainment into the free troposphere. For the golden flights, FCDP number concentration ranged from $0.0^{+0.1}_{-0.0}$ cm$^{-3}$ in the BL

and FT (< 200 km) to $0.4^{+0.2}_{-0.1}$ cm$^{-3}$ in the BL and $0.0^{+0.1}_{-0.0}$ cm$^{-3}$ in the FT (> 900 km). Note that zero values do not preclude the

possibility of there being particles between 3 – 50 μm since the FCDP cannot measure concentrations below 0.05 cm$^{-3}$. For

RFs 156-157 within the BL, FCDP number concentrations close to Bermuda increased to $1.3^{+0.9}_{-0.6}$ cm$^{-3}$, but no clear trend existed

for the last four flights.

Reid et al. (2001) observed higher coarse aerosol concentration in the BL with offshore distance (up to just over 40 km

offshore) by the North Carolina Outer Banks area in conditions of offshore flow. At least one other study examining

supermicron particle number concentration with offshore distance reported that there was an increase with offshore distance

off the California coast for certain conditions (Schlosser et al., 2020). These and other past studies pointed to the great

complexities associated with marine supermicron aerosol concentrations due to BL dynamics, dependence of sea salt

generation of multiple factors, and measurement technique limitations (Reid et al., 2006; Porter and Clarke, 1997; Reid et al.,

2001; Smirnov et al., 2003), which emphasizes the difficulty of associating supermicron aerosol concentrations with any one

specific governing factor like wind speed. Wind speed is shown for context along the Falcon flight tracks in Fig. S6. It is not



possible to draw any clear relationship between winds and supermicron aerosol, and that is outside the scope of this study. It is also important to consider the fetch of the air mass. If the flow is parallel to the coast (as it is for RFs 159, 160, and 179), we would not expect distance offshore to matter much.

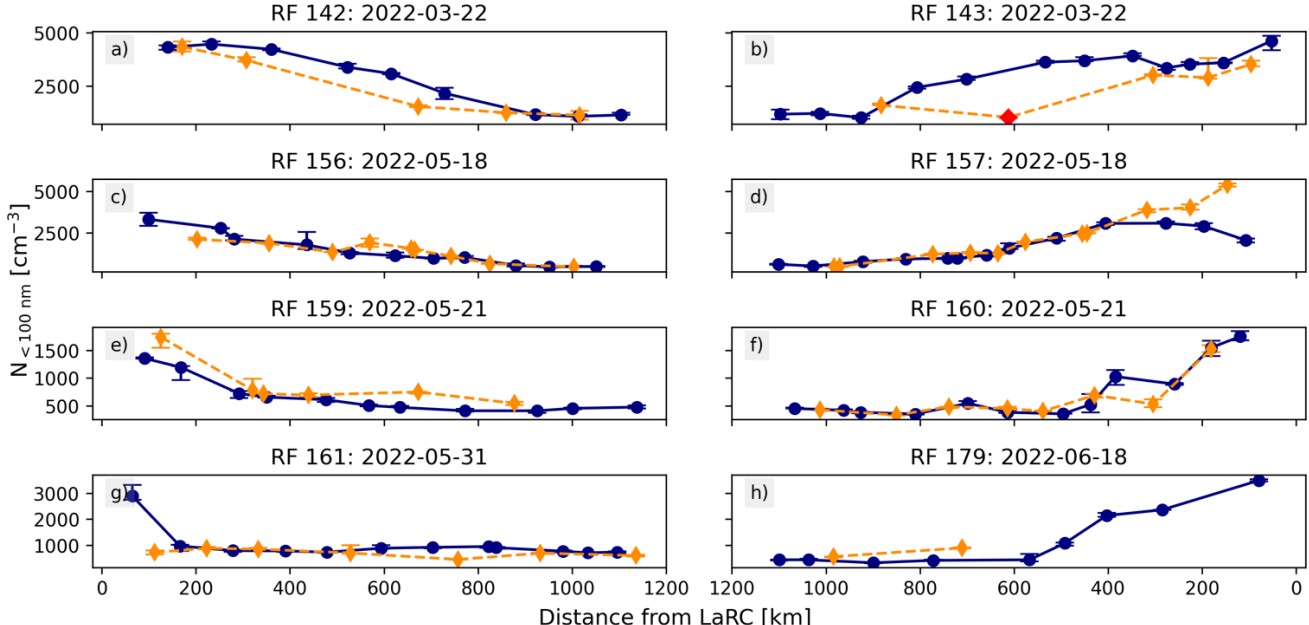


**Figure 6: Particle number concentrations ($D_p < 100$ nm) versus offshore distance relative to LaRC. Blue and orange represent the BL and FT, respectively. Smoke layer median value shown in red for RF 143. Markers represent median values for 15-minute intervals and whiskers are 25th/75th percentiles.**



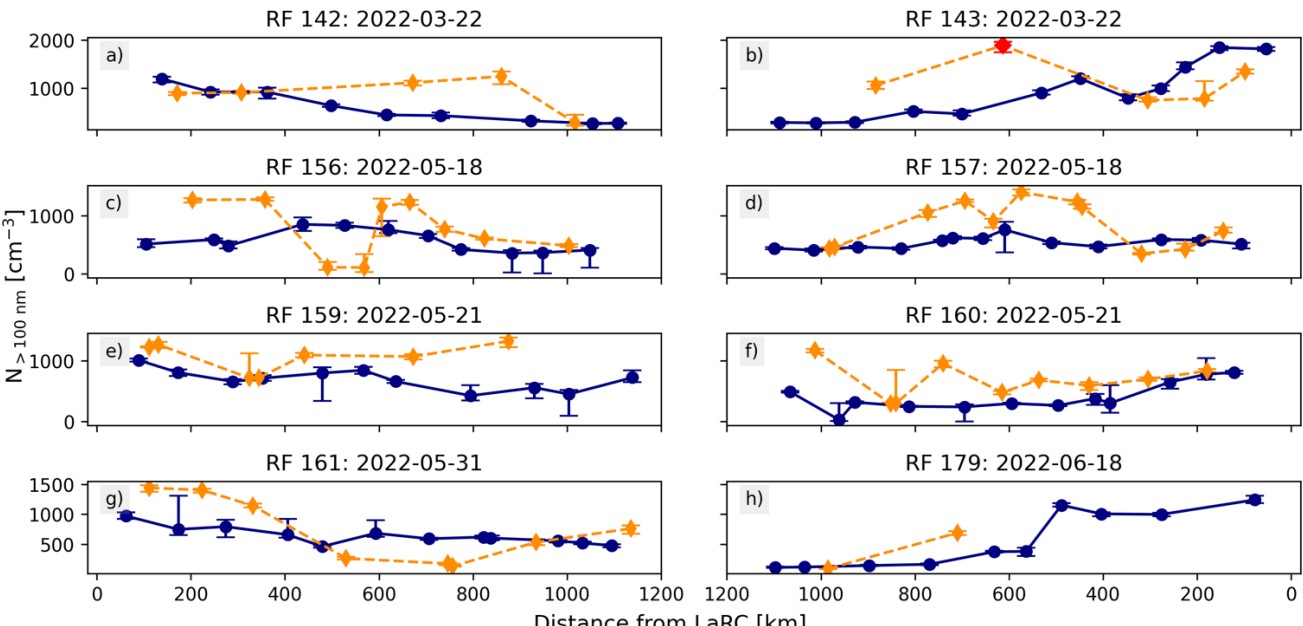

**Figure 7: Same as Figure 6 but for D$_p$ > 100 nm.**

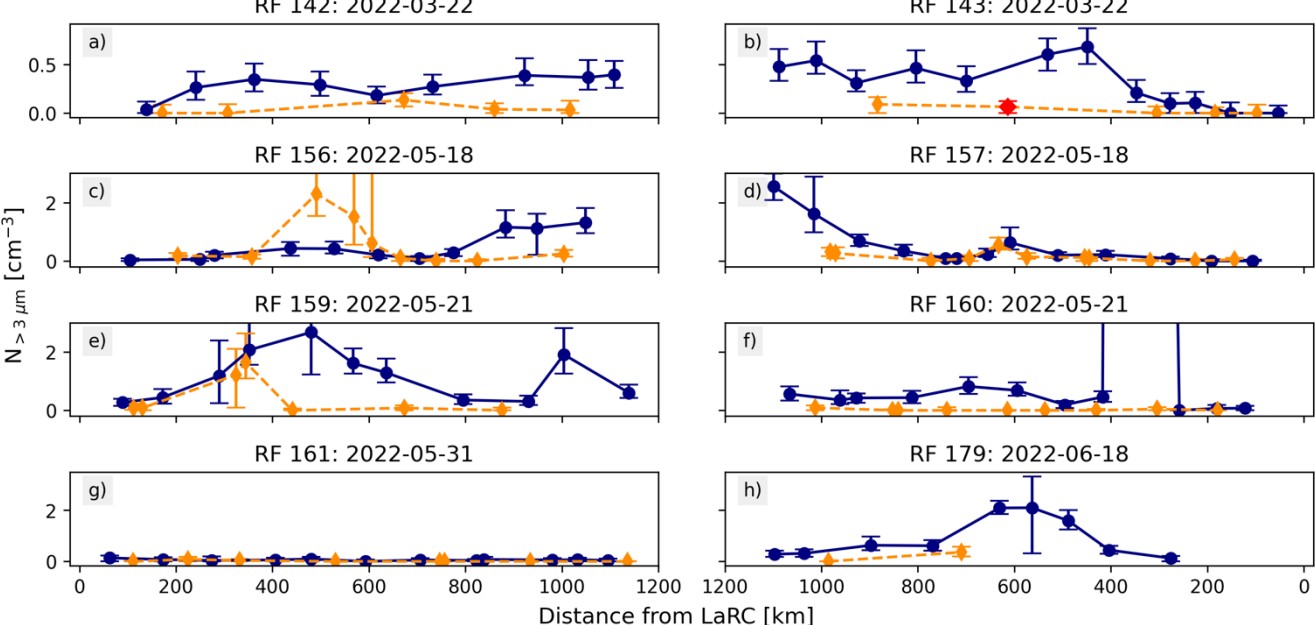

**Figure 8: Same as Figure 6 but for D$_p$ > 3 μm.**





### 3.4.2 Number size distributions

Figures 9 (BL) and S7 (FT) show particle size distributions ranging from 3 to 5000 nm segmented into four offshore distance
ranges relative to LaRC: <300 km, 300 – 550 km, 550 – 800 km, and >800 km. The data were fit with a two-mode, lognormal
size distribution, with parameter values for total number concentration (N), geometric mean diameter ($D_{p,g}$), and geometric
standard deviation ($\sigma_g$) of the fit for modes 1 and 2 shown in Tables S1-S3. Figure S8 further shows how these three parameters
varied for the two modes between flight days. Across all flights, mode 1 represents the Aiken mode ($D_{p,g} = 43^{+16}_{-4}$ nm and $58^{+20}_{-22}$
nm in the BL layer and FT, respectively) while mode 2 is representative of the accumulation mode ($D_{p,g} = 112^{+66}_{-13}$ nm and
$143^{+48}_{-34}$ nm in the BL layer and FT, respectively). The Hoppel Minimum, the dip between the Aitken and accumulation modes
occurring around the 50 to 80 nm range indicating cloud processing (Hoppel et al., 1986), is more prominent with distance
from LaRC in BL measurements. In the FT, the Hoppel Minimum is not clearly evident, owing to the likelihood that the
particles in the BL and FT are unrelated with different histories.

Figure S8 shows that the number concentration of the BL Aitken mode decreases with offshore distance for 4 of the 5 flight
days. There was also a decrease in the BL Aitken mode $D_{p,g}$ with offshore distance for 3 of the 5 flight days overall, with the
reduction most pronounced for the golden flights (RF 142-143) where the BL Aitken mode $D_{p,g}$ decreased by 36% (58%) from
LaRC to Bermuda for the morning (afternoon) flight. No clear trends exist for N and $D_{p,g}$ in the FT for both modes and for the
BL accumulation mode measurements. The geometric standard deviation also showed no clear trends in the BL and FT.

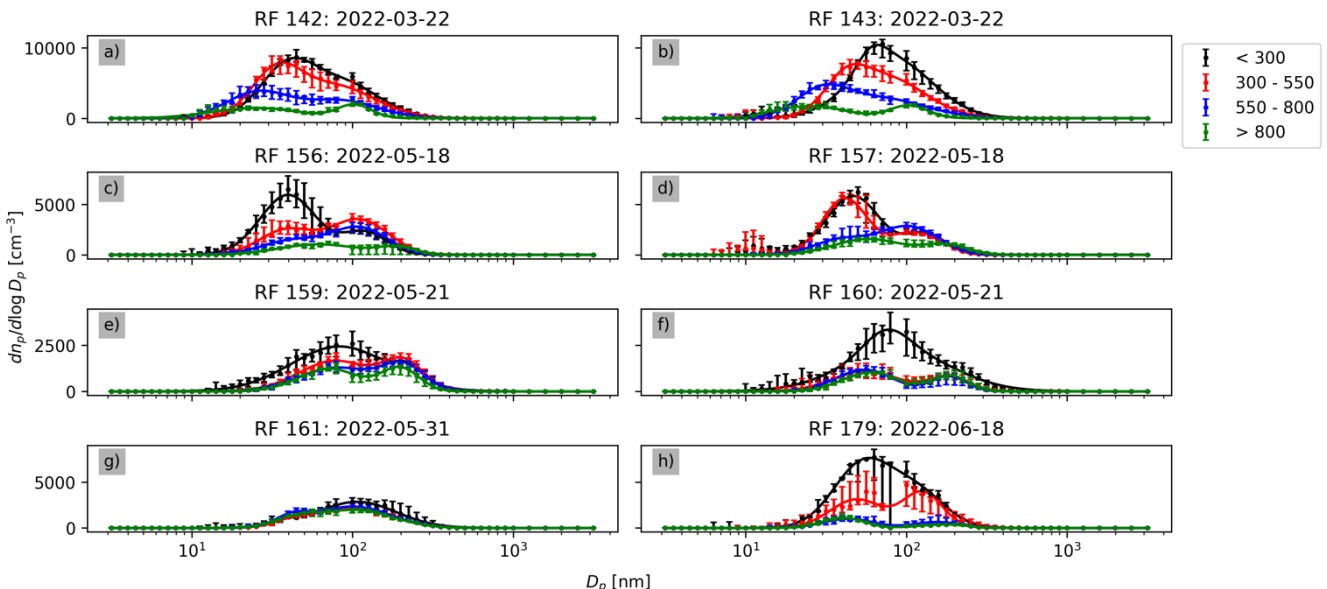

**Figure 9: Median particle size number concentration distributions in the boundary layer with best-fit models segmented by four offshore distance ranges (in km) relative to LaRC. 25th and 75th percentiles are signified with error bars.**



### 3.4.3 New particle formation

The number concentration ratio of particles with diameters > 3 nm to particles with diameters > 10 nm ($N_{>3 nm}/N_{>10 nm}$) has traditionally been used as an indicator of new particle formation (NPF) occurring in the time leading up to the measurement
(e.g., Clarke, 1993; Covert et al., 1992; Corral et al., 2022; Zheng et al., 2021). Newly formed particles typically have a diameter around 1 nm (Pirjola et al., 2000), which is too small to measure with the Falcon instruments. Thus, growth of newly formed particles must occur before they can be directly measured in ACTIVATE.

Figure 11 displays $N_{>3 nm}/N_{>10 nm}$ for all transit flights when data are available. The ratio remains relatively constant with offshore distance and is consistently higher (albeit slightly) in the FT ($1.327^{+0.039}_{-0.033}$) compared to the BL ($1.283^{+0.033}_{-0.031}$). In terms
of absolute number concentration between 3 and 10 nm, Fig. S9 shows a general offshore decline for all flights without much difference between the BL and FT. In previous studies, concentrations of particles smaller than 20 nm are low within the pristine MBL, indicating NPF is infrequent in the marine boundary layer (Raes et al., 2002; Heintzenberg et al., 2004; Saltzman, 2009). Modeling supports the lack of new particle formation in typical clean non-polar MBL environments (Pirjola et al., 2000). In contrast, the longer residence times in the free troposphere and generally lower pre-existing surface area to
scavenge condensable gases can promote NPF (O'Dowd et al., 1996). New particle formation has been observed in the upper remote marine boundary layer after the passage of a cold front during the Aerosol and Cloud Experiments in the Eastern North Atlantic (ACE–ENA) campaign (Zheng et al., 2021) and high levels of nucleation mode particles were observed in the marine boundary layer as part of the NASA North Atlantic Aerosols and Marine Ecosystems Study (NAAMES) (Gallo et al., 2023). The results from this study suggest there is slightly more prevalence of newly formed particles in the FT, enhancing particle
concentrations between 3 and 10 nm compared to the boundary layer. This is important since entrainment from the free troposphere plays a significant role in influencing the MBL budget for aerosol and CCN number concentration, whereas particle mass is dominated by sea salt (David et al., 1999; Tornow et al., 2022). The results of this analysis are consistent with previous work for the region showing higher $N_{>3 nm}/N_{>10 nm}$ above the BL due to at least some combination of meteorological conditions (cold and dry air), reduced aerosol surface area, and high precursor vapor levels from continental outflow (Corral
et al., 2022). The uniformity of $N_{>3 nm}/N_{>10 nm}$ values with offshore distance reflects some balance between influential factors including proximity to precursor vapor sources, meteorological conditions, and available aerosol surface area to serve as a condensation/coagulation sink for precursor vapors (Pirjola et al., 2000). We note that there was no inverse relationship between aerosol surface area and either $N_{>3 nm} - N_{>10 nm}$ (Fig. S10) or $N_{>3 nm}/N_{>10 nm}$ (not shown).




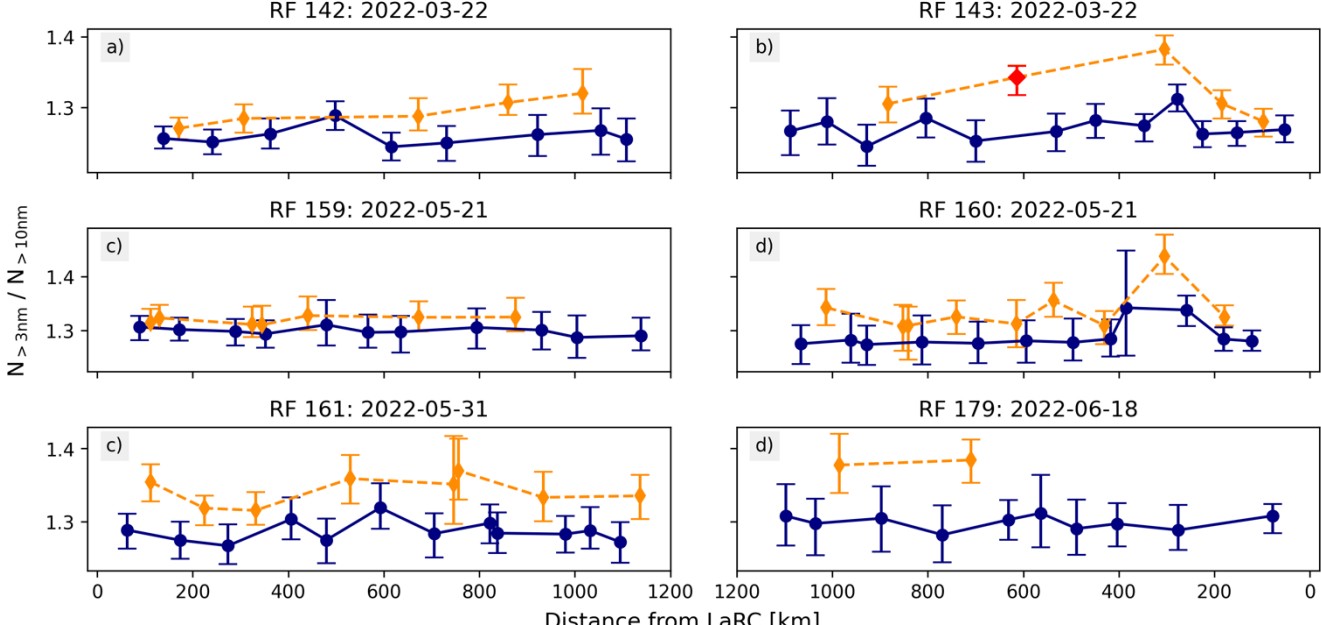

**3.5 Optical properties**

Submicron scattering (Fig. 12) and absorption coefficients for particles with D$_P$ < 5 μm (Fig. 13) showed a slight decrease
with offshore distance in the BL for the golden flights. Near LaRC (< 200 km), the BL scattering coefficient was 26.8$^{+4.8}_{-7.3}$ Mm$^{-1}$
and the absorption coefficient was 2.8$^{+0.5}_{-0.5}$ Mm$^{-1}$. Near Bermuda (> 900 km from VA), BL scattering and absorption
coefficients decreased to 9.7$^{+1.7}_{-1.4}$ Mm$^{-1}$ (−64%) and 0.3$^{+0.6}_{-0.6}$ Mm$^{-1}$ (−89%), respectively. The golden flight scattering and
absorption coefficients were more variable in the FT. The opposite trend was observed for RFs 156-157, with scattering
increasing with offshore distance in the BL and FT. Lower particle surface area concentrations were observed near the coast
on this day, which may have led to the lower scattering coefficient values near the coast. That day (18 May) coincided with
the highest APT values along the flight track, including close to the VA coast (Fig. 1), which may have played a role as well
in the reduced scattering. The scattering coefficient within the FT showed no clear trend across flights. The absorption
coefficient was near constant for RFs 156-157 due to the potential for wet scavenging of sampled air masses, whereas it showed
a decrease with offshore distance for all the other flights consistent with black carbon being the most significant contributor
to aerosol absorption and it not having a secondary source.




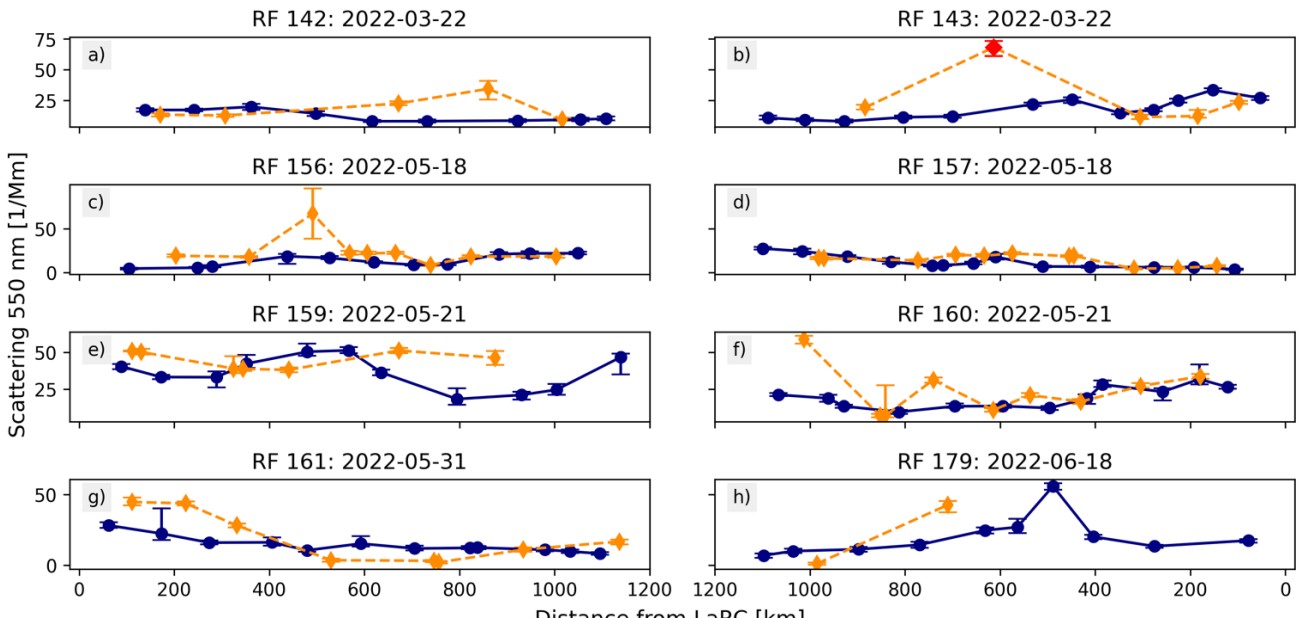

**Figure 11: Submicron dry scattering coefficient at 550 nm versus offshore distance relative to LaRC. Blue and orange represent the BL and FT, respectively. Smoke layer median value shown in red for RF 143. Markers represent median values for 15-minute intervals and whiskers are 25th/75th percentiles.**

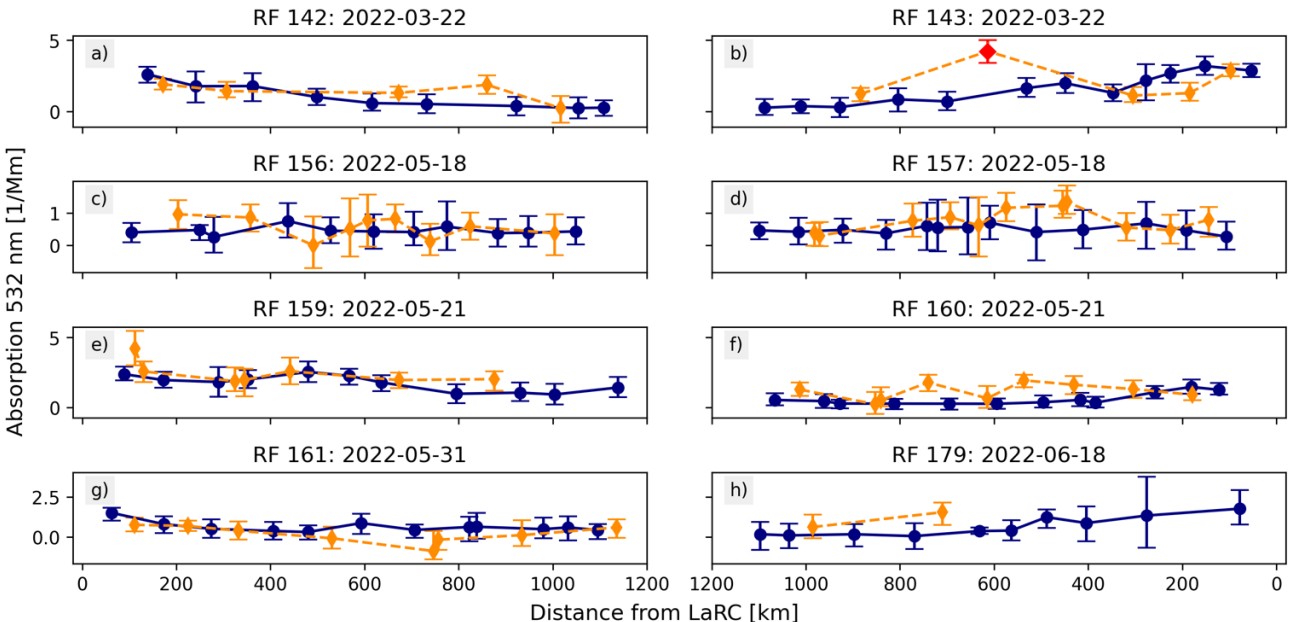

**Figure 12: Same as Fig. 11 but for absorption coefficient at 532 nm.**



## 3.6 Aerosol chemistry

A previous study focused on the northwest Atlantic showed using the first two years of ACTIVATE AMS data that sulfate
and organics dominated the submicrometer non-refractory aerosol mass, and that organics showed more of an offshore gradient
while sulfate was more spatially homogeneous (Dadashazar et al., 2022a). That study did not have access to transit flight data
to Bermuda as these only occurred in 2022 flights, which is the focus of this discussion. Speciated aerosol mass concentrations
from the AMS are discussed for the BL (Fig. 13) and FT (Fig. 14). As expected, particle mass was dominated by organics and
sulfate across flights. BL data from the second golden flight is unavailable near Bermuda. Still, RF 142 results show a clear
decrease in aerosol mass with distance and a transition from more organic-rich particles near the continent to sulfate-rich
particles near Bermuda. For the golden flights, total submicron non-refractory mass decreased from $10.9^{+1.0}_{-2.6}$ µg m$^{-3}$ near LaRC
(< 200 km) to $1.4^{+0.2}_{-0.1}$ µg m$^{-3}$ near Bermuda (> 900 km), corresponding to an 87% decrease. A similar, but more non-linear
evolution in particle mass fraction is observed for the other research flights. Unlike the other flights, the decrease in organic
mass fraction in the BL for the high APT flights (RFs 156-157 on 18 May 2023) resulted from an increase in sulfates near
Bermuda rather than a decline in organic mass concentration. For the golden flights, organic mass concentration decreased
from $8.0^{+0.7}_{-1.3}$ µg m$^{-3}$ near LaRC (< 200 km) to $0.5^{+0.2}_{-0.1}$ µg m$^{-3}$ near Bermuda (> 900 km), coinciding with a 94% decrease.
Meanwhile, sulfate mass concentration remained relatively more steady between $1.0^{+0.1}_{-0.2}$ µg m$^{-3}$ to $0.7^{+0.1}_{-0.0}$ µg m$^{-3}$ at the same
locations, equating to just a 30% decrease. For RFs 156-157, organics remained more constant between LaRC and Bermuda
($1.9^{+0.4}_{-0.2}$ to $1.4^{+0.3}_{-0.1}$ µg m$^{-3}$) while there was an 189% increase in sulfate mass concentration ($0.9^{+0.1}_{-0.2}$ to $2.6^{+0.2}_{-0.2}$ µg m$^{-3}$).
Generally, higher aerosol mass concentrations were measured in the FT. A less pronounced transition from organic to sulfate
particles was observed in RF 142, but organic particles were the dominant FT species across all flights. There were no clear
trends in the spatial evolution of aerosol species in the FT for the other flights.



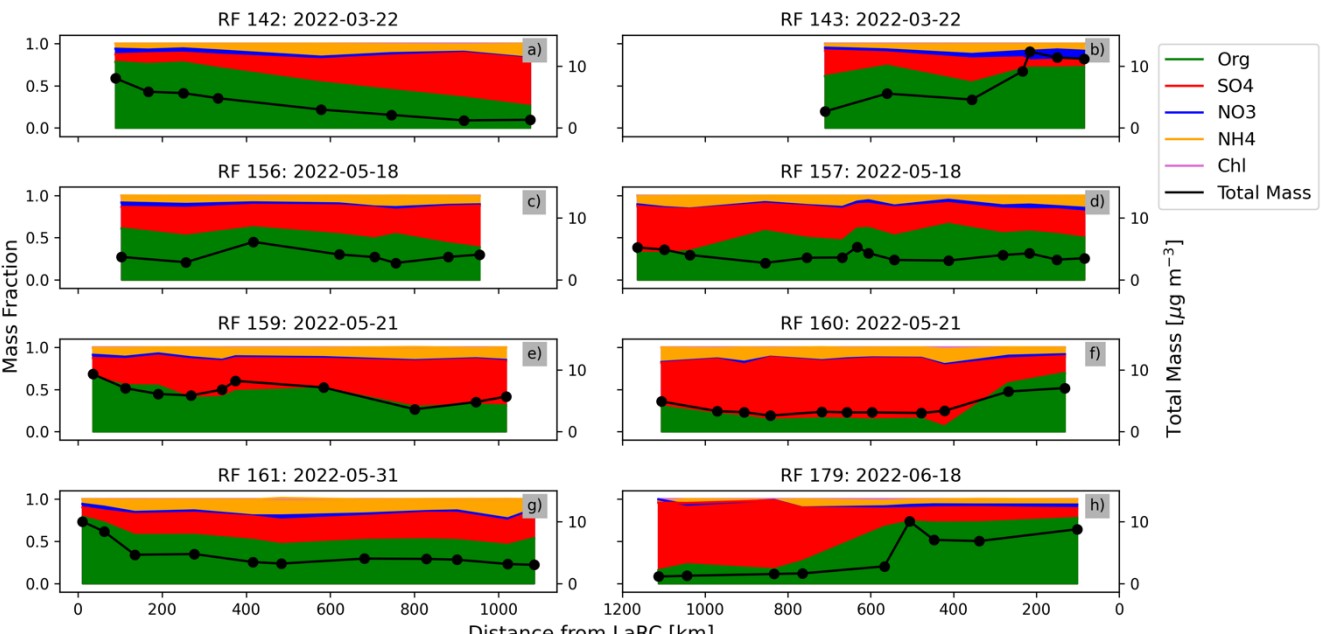

**Figure 13: Speciated mass fractions for submicron non-refractory aerosol components as measured by the AMS in the BL. Total mass concentration displayed with black line. Data averaged across 10-minute time intervals.**

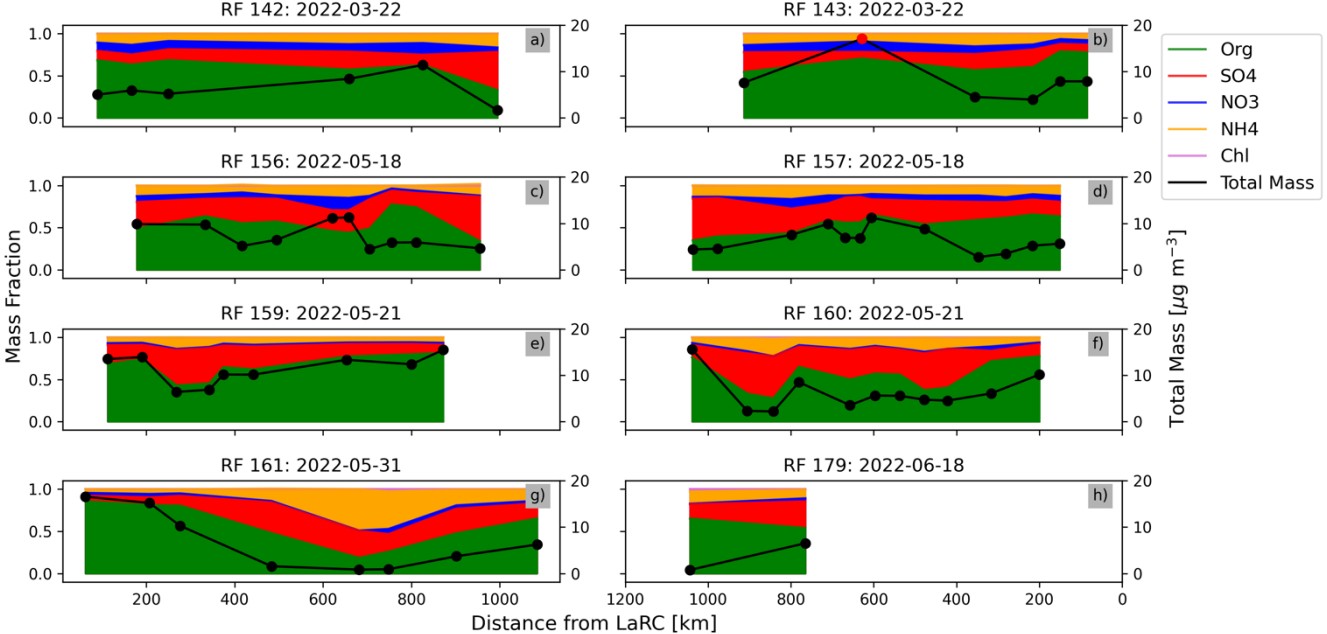

**Figure 14: Same as Fig. 13 but for measurements in FT. Smoke layer mean total AMS non-refractory aerosol mass shown in red for RF 143.**





By looking at the concentration of CO compared to background levels, the anthropogenic influence of an air mass can be
estimated. Using this method, AMS mass concentrations can be correlated with anthropogenic influence. As outlined in
Dadashazar et al. (2021), CO is sourced from North American pollution (Corral et al., 2021) with key traits being both having
a lifetime of about a month (Weinstock, 1969) and exhibiting a high resilience to wet savaging. $\Delta CO$ is calculated by
subtracting the flight-specific $5^{th}$ percentile CO concentration in the BL from any given CO data value. Figure 15 displays
total AMS mass versus $\Delta CO$ for BL measurements. If mixing between polluted and clean air masses is the prominent control
on mass concentrations, then one would expect a linear relationship and with a constant positive slope. This is observed clearly
for the golden flights and on 21 May, although $\Delta CO$ is not correlated to offshore distance on 21 May compared to the golden
flights. The negative slope observed on 18 May suggests that the air mass proportionally gained aerosol mass during transport,
although the nonlinearity suggests additional controls beyond simple mixing. Mass concentrations were lower by a factor of
approximately three near the VA coast for 18 May compared to the golden flights, analogous to relative differences in particle
number concentrations (Sect. 3.4). Wet scavenging due to higher APT values is presumed to have led to the lower mass
concentrations on 18 May (Fig. 14, B). Near Bermuda, AMS mass for the 18 May flights is twice as large as the golden flights.
As we saw from the HYSPLIT modeling, on 18 May 2022, the air mass shifted to being of marine origin near Bermuda. Thus,
the sulfate enhancement near Bermuda could result from marine biogenic emissions of dimethyl sulfide (DMS), which can
produce sulfate after secondary reactions (Seinfeld and Pandis, 2016). The results for the other flights are more variable
because there was a mix of air masses intercepted along the flight path (Fig. 2, E-H).

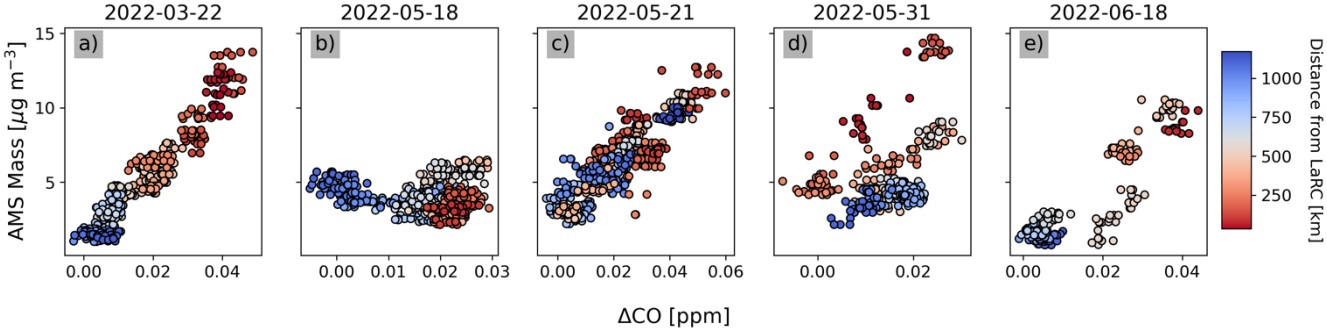

**Figure 15: BL total AMS submicron non-refractory mass versus $\Delta CO$ colored by distance relative to LaRC.**

During long-range transport, particles can oxidize at differing rates in the BL and FT (Schum et al., 2018). Utilizing transit
flight data, the change in the oxidation state of particles can potentially be inferred spatially by looking at the fraction of
organic mass containing mixed hydrocarbons ($f_{43}$) compared to oxidized hydrocarbons ($f_{44}$). Figure 16 compares $f_{44}$ versus $f_{43}$
in the BL, which is a commonly used method to determine if particles have undergone extensive atmospheric aging to yield
low-volatility oxygenated organic species (i.e., points converging at the top left of the triangle) (Ng et al., 2010). For the golden
flights, there does not seem to be a clear trend with data farther offshore moving more to the upper left of the triangle. For





flights on 18 May 2022 and 31 May 2022, the points do appear to move towards the top left with offshore distance. The 31

May flights (Fig. 16, D) captured continental outflow but at lower velocities than the golden flights based on distance covered by trajectories for similar time ranges. The lower velocity flow could have given more time for these particles to oxidize although we are cognizant that it is hard to prove any causal relationships with this analysis. While the 18 May flights (Fig. 16, B) experienced lower velocity flow near Bermuda compared to the golden flights, there was also an air mass shift at that location which could have influenced the $f_{43}$ and $f_{44}$ changes. In the FT (Fig. 16, A-F), there a far fewer AMS measurements

but again there is evidence of aging on 18 May 2022 and 31 May 2022.

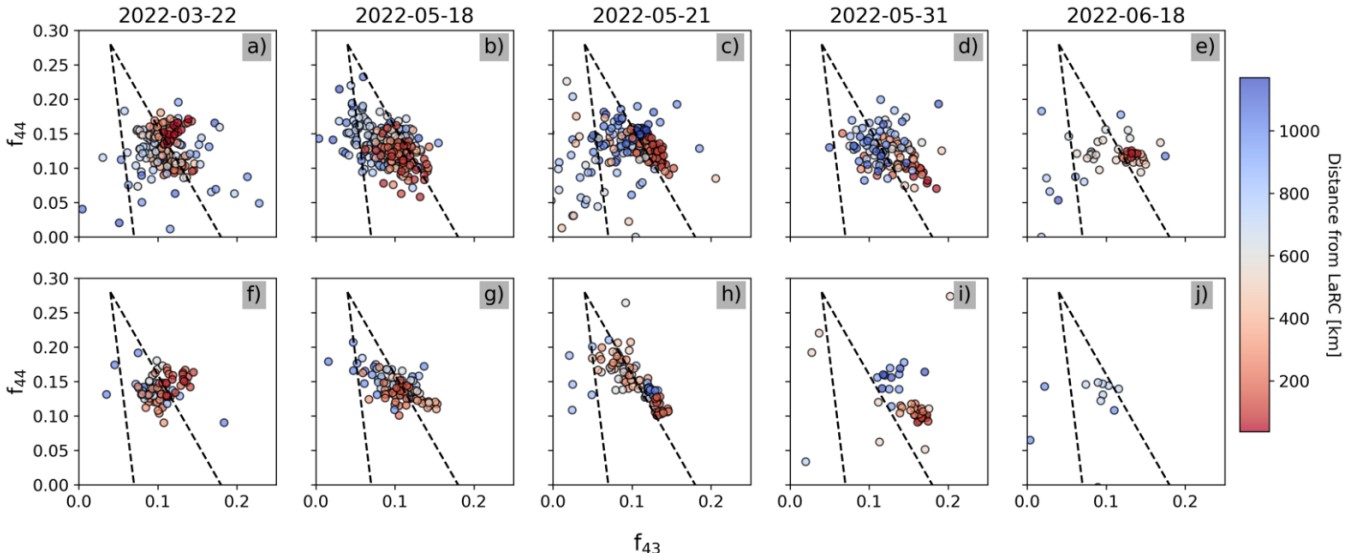

**Figure 16: Scatter plot of $f_{44}$ versus $f_{43}$ colored by distance from LaRC for (a-e) boundary layer and (f-j) free tropospheric data. The dashed triangle is added for scale in reference to previous studies (e.g., Ng et al., 2010), which utilize this type of plot.**

### 3.6.1 Particle hygroscopicity

Understanding aerosol hygroscopicity is important for characterizing global climate because the water uptake of particles is relevant for optical properties and droplet activation behavior (Huang et al., 2022). Sea salt is a highly hygroscopic aerosol type while insoluble organic compounds exhibit low hygroscopicity (Schill et al., 2015). Continental and anthropogenic emissions typically have lower hygroscopicity (Xu et al., 2020), suggesting hygroscopicity would increase with distance from the coast. Figure 18 generally shows an increase in hygroscopicity as organic mass fraction decreases offshore in the BL. Note

these measurements of hygroscopicity correspond to fine-mode aerosol and not the total (fine and coarse mode) aerosol. We have seen large differences between nephelometer measurements and the hygroscopicity derived from HSRL-2 and dropsonde data, which typically have significantly larger values (i.e., subject of forthcoming work). The range of f(RH) values was relatively consistent across flights. In the BL for the golden flights, f(RH) increased 25% from $1.2^{+0.1}_{-0.1}$ within 200 km of LaRC



to $1.5^{+0.3}_{-0.2}$ near Bermuda for measurements > 900 km offshore. In the FT for the golden flight (Fig. 18, F), f(RH) showed only

a 7% increase ($1.2^{+0.1}_{-0.1}$ to $1.3^{+0.3}_{-0.2}$) but there was higher variability and fewer measurements made.

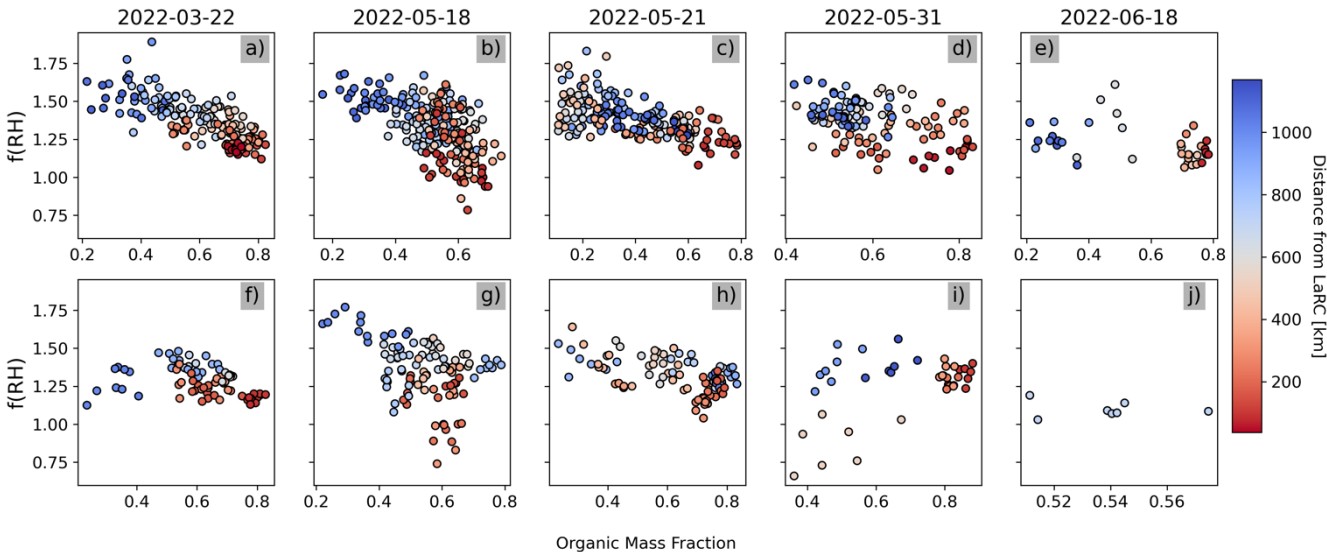

**Figure 17: Scatterplot of f(RH) versus organic mass fraction for (a-e) boundary layer and (f-j) free tropospheric data, all colored by distance from LaRC.**

**3.7 Cloud condensation nuclei**

For the golden flights and RF 179, CCN was measured at a constant supersaturation (*s*) of 0.37%, which is the focus of this analysis, unlike other flights that had varying supersaturations, making it harder to draw conclusions from spatial trend analysis. Figure 18 displays CCN concentration as a function of offshore distance from LaRC. The golden flights and RF 179 show a decline in CCN with offshore distance in the BL but a less clear trend in the FT. For the golden flights within the BL,

values were $3000^{+210}_{-320}$ cm$^{-3}$ within 200 km of LaRC and $480^{+60}_{-40}$ cm$^{-3}$ near Bermuda for distances over 900 km from LaRC, corresponding to an 84% decrease.



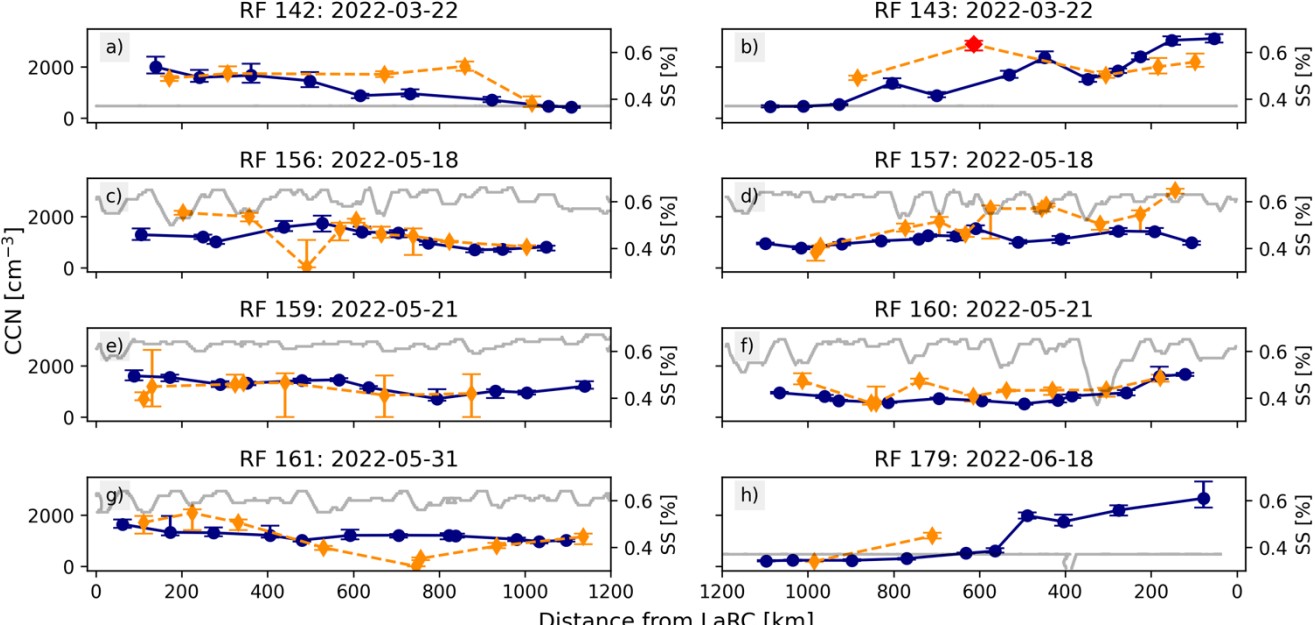

**Figure 18: CCN Number concentration versus distance from LaRC. Gray line displays supersaturation of the CCN Spectrometer. Blue and orange represent the BL and FT, respectively. Smoke layer median value shown in red for RF 143. Markers represent median values for 15-minute intervals and whiskers are 25th/75th percentiles.**

### 3.8 Smoke layer

On the day of the golden flights (RF 142-143 on 22 March 2022), there existed a layer aloft as discussed in Sect. 3.2 and shown in the HSRL-2 backscatter curtain in Fig. 3. Based on data collected during a single FT leg about 610 km from LaRC, which directly intersected this layer, we can see trace gas and aerosol signatures of smoke. As shown in Fig. 5, the $\Delta CH_4/\Delta CO$ ratio becomes <0.3 within the smoke leg, indicating a biomass-burning source (Mauzerall et al., 1998). Figures with distance from LaRC on the x-axis throughout this paper mark measurements in this opportune layer in red. Enhancements in the smoke layer occurred in absorption and submicron scattering coefficients, as well as $CH_4$, CO, and $CO_2$ concentrations. No pronounced increase was observed for $N_{<100nm}$, $N_{>3\mu m}$, $N_{>3nm}/N_{>10nm}$, $N_{>3nm} - N_{>10nm}$, f(RH), or $O_3$. AMS mass concentrations show elevated organics, nitrate, and ammonium, characteristic of smoke. m/z 60 corresponds to $C_2H_4O_2^+$, a tracer of biomass burning. In the smoke leg, m/z 60 concentrations were enhanced.

The geometric mean diameter of fresh smoke is typically in the 100–160 nm range with particle aging causing these particles to grow to the 120–230 nm range (Reid et al., 2005). Other work has shown smoke particles in the western U.S. having diameters mainly above 100 nm (Mardi et al., 2018). The particle size distribution observations line up with these ranges. Enhancements in the smoke layer were observed in $N_{>100\ nm}$ concentrations but not in the $N_{<100\ nm}$ concentrations. $D_{p,g}$ values for the two-mode lognormal fit were 96 and 208 nm. The CCN concentrations were also elevated in the smoke layer. Figure



19 shows the direct intersection with the smoke layer in blue and two intersections with the layer near the transition into the boundary layer in black and red. The boundary layer measurements near Bermuda (>900 km from LaRC) are shown in green. The smoke layer concentrations are highest in the direct intersection and $D_g$ of each mode is larger than the intersection near the transition into the boundary layer. The smoke leg lacks an Aitken mode which occurs at a geometric mean diameter of 36

nm in the transition leg and at 26 nm in the BL near Bermuda. There still exists an accumulation mode with a $D_{p,g}$ of ~120 nm in the transition layer and 104 nm in the BL near Bermuda. As the layer further diffuses, the smoke can enter the boundary layer, affecting CCN concentrations (Colarco et al., 2004; Zheng et al., 2020). However, CCN enhancements appeared to be constrained to the layer. These results are significant with implications for smoke layers that are common off the U.S. East Coast (e.g., Mardi et al., 2021).

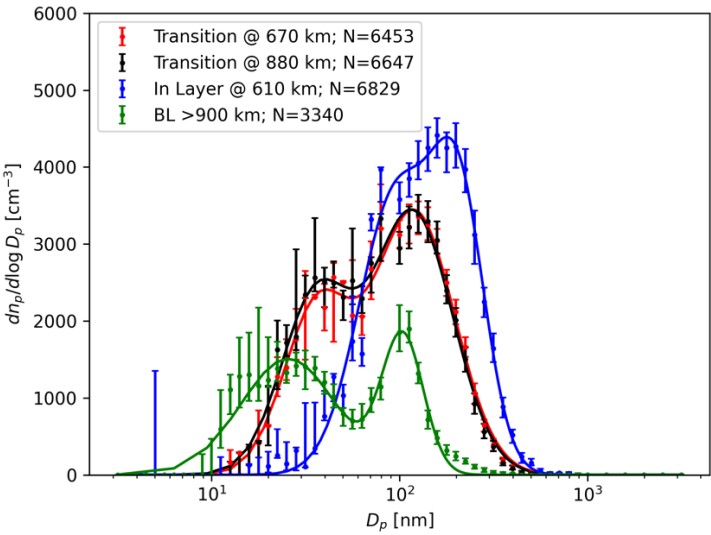


**Figure 19: Number size distribution of particles for the golden flights (RF 142-143 on 22 March 2022) within the smoke layer (blue), in the transition between the layer and the BL (red & black), and in the remote BL (green); locations of the layer are shown in Fig. 3. 25th and 75th percentiles are signified with error bars. The two-mode lognormal best fits are shown with solid lines. The distance from LaRC of each measurement is indicated in the legend along with the total number concentration in cm⁻³ of the best-fit model.**

## 4 Conclusion


This study utilized data from eight transit flights between LaRC and Bermuda from the ACTIVATE campaign to bridge aerosol particle and gas properties between these two regions. After identifying the predominant atmospheric circulation patterns for each of the eight transit flights, the first day of flights was shown to closely follow the stream of continental outflow. The other flight days had a mix of continental and marine influences. Using the first day of flights, called the "golden

flights", as an ideal case and the other days for comparison, we identified gas and aerosol particle properties that changed or remained constant with distance from LaRC.

$\Delta CH_4/\Delta CO$ values generally indicated urban/ocean emissions and not biomass burning as the dominant gas and particle source. The exception was $\Delta CH_4/\Delta CO$ measurements made within a smoke layer, identified in the golden flights using $CH_4$, $CO$, and $O_3$ concentrations as well as aerosol particle scattering, size distribution, and chemistry data. In the boundary layer, $CH_4$, $CO$,

and $CO_2$ concentrations decreased with offshore distance. Boundary layer particle $N_{<100\ nm}$, $N_{>100\ nm}$, Aitken mode $D_{p,g}$, Aitken mode N, and organic mass fraction also decreased offshore. In the free troposphere, $N_{<100\ nm}$ also exhibited an offshore decrease. Within the boundary layer, trace gas and particle number concentrations leveled off around ~900 km from LaRC for the golden flights. BL sulfate mass fraction and f(RH) increased offshore while $N_{>3\ nm}/\ N_{>10\ nm}$ was constant but slightly higher in the FT than the BL.

These results provide important case studies utilizing a wide range of measurements to understand the evolution of gas and aerosol particle properties off the U.S. East Coast. Characterizing gas and particles along with the scale of the transition between continental and marine environments will better assist modelers in parameterizing the complex array of dynamic and chemical atmospheric processes. These transit flights give us a narrow slice of the interaction between continental and marine environments specific to the northwest Atlantic. The results demonstrate the many atmospheric variables that exhibit offshore

gradients in the study region, with trends generally more prominent within the BL.

**Data availability**

The ACTIVATE dataset is publicly accessible at https://doi.org/10.5067/SUBORBITAL/ACTIVATE/DATA001 (Activate Science Team, 2020).

**Author contributions**

YC, ECC, JPD, GSD, MAF, RAF, JWH, SK, RHM, TJS, MAS, KLT, CV, ELW, and LDZ collected and/or prepared the data. CS and MRH conducted data analysis. CS, FG, and AS conducted data interpretation. CS and AS prepared the manuscript with editing from all co-authors.

**Competing interests**

At least one of the (co-)authors is a member of the editorial board of Atmospheric Chemistry and Physics.



## Disclaimer

Publisher's note: Copernicus Publications remains neutral with regard to jurisdictional claims in published maps and institutional affiliations.

## Acknowledgments

The authors acknowledge the pilots and aircraft maintenance personnel of NASA Langley Research Services Directorate for conducting ACTIVATE flights and all others who were involved in executing the ACTIVATE campaign. The authors gratefully acknowledge the NOAA Air Resources Laboratory (ARL) for the provision of the HYSPLIT transport and dispersion model and READY website (https://www.ready.noaa.gov) used in this publication.

## Financial support

ACTIVATE is a NASA Earth Venture Suborbital-3 (EVS-3) investigation funded by NASA's Earth Science Division and managed through the Earth System Science Pathfinder Program Office. University of Arizona investigators were supported by NASA grant no. 80NSSC19K0442 and Office of Naval Research grant no. N00014-21-1-2115. CV and SK were funded by DFG SPP-1294 HALO under project no. 522359172 and by European Union under grant no. 101101999 and by SESAR JU CICONIA.

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
