# Peer review of "Bridging Gas and Aerosol Properties between Northeast U.S. and Bermuda: Analysis of Eight Transit Flights"

_EGUsphere, 2024_

## Author Comment (AC1)

**Manuscript ID: egusphere-2024-926**

We would like to thank the reviewers for providing clear and valuable feedback on our manuscript. We have addressed your feedback in blue beneath each comment.

**Reviewer 1**

**Reviewer Summary**

In this study the authors describe the results of 8 transit flights between Langley Research Center in Hampton VA and Bermuda as part of the ACTIVATE study in the northwestern Atlantic Ocean in 2022. Two planes flying at different altitudes tracked a path to Bermuda and back, with many on-board instruments to investigate different components of the gas and aerosol composition along this path. This study provides a great deal of insight into the characteristics of gas and aerosol composition within the marine boundary layer and the free troposphere. Overall, this is a well written and organized paper that provides significant value to the atmospheric science community and should be published after consideration of a few minor questions and comments.

**Comments**

Comment #1, Pg 10 Lines 210-221:

In this section you are discussing how sea salt can be mis-identified as dust-mix at RH < 75%. You state: "we presume this is the case during these flights as the RH in the BL was usually below 75% based on Falcon…", so does that imply that you are assuming that the "dusty mix" (for at least the first four flights) is actually "dry" sea salt? And is the lack of "dusty-mix" on wetter days an argument in favor of the dusty-mix on the "dry" days being sea salt? Perhaps this section could be re-worded a bit to make it a little more clear what is happening and what conclusions you are drawing from these observations.

Response: Yes, that is correct. We slightly reworded that sentence for clarity: "We presume a misclassification of dry sea salt as "dusty mix" occurred during the golden flights as the RH in the BL was usually below 75% based on Falcon (Fig. S3) and King Air dropsonde measurements (Fig. S4), especially on the morning flight."

Comment #2, Pg 11 Lines 227-229

While there is definitely a decrease in the CH4 and CO concentrations from LARC to Bermuda, it doesn't seem that large in either case. How do these numbers compare to a more "background" concentration of these gases in this region? Is the number observed at Bermuda representative of a clean background, or is it still elevated?

Response: Thank you for raising this question. We agree it is important to compare these values to ground-based measurements to identify if we are seeing enhancements beyond background levels. We have added a few ground-based measurements for comparison in response to

Comment #3 (see below). Both $CH_4$ and CO aircraft measurements are higher than ground-based measurements near Bermuda.

Comment #3 Pg 11 Lines 229-230

This is somewhat related to previous comment, but you mention that the concentrations "level off" around 900 km offshore, what concentration do they level off to? Is it the concentration observed at Bermuda?

Response: Concentrations level off to the values stated a couple of lines above: "For the golden flights, $CH_4$ (CO) concentrations decreased from $2.056^{+0.005}_{-0.006}$ $(0.165^{+0.004}_{-0.003})$ within 200 km from NASA Langley Research Center (LaRC) to $1.989^{+0.001}_{-0.001}$ $(0.131^{+0.003}_{-0.003})$ ppm at >900 km from LaRC."

For more clarity, we revised and made additions to the following text: "It is noteworthy that the gas concentrations for the golden flights seemed to level off approximately ~900 km from LaRC to the values reported above. This is coincident with past analysis based on MERRA-2 reanalysis data showing a distinct offshore gradient in tropospheric CO from the U.S. East Coast towards Bermuda, with most of the decline within the first few hundred kilometers offshore (Corral et al., 2021). Our measured aircraft values near Bermuda are slightly higher than ground-based measurements in Bermuda. Between 2010 – 2014, ground-based measurements of $CH_4$ in Bermuda ranged from 1.8 to 1.9 ppm, peaking around January and reaching a minimum around June (Turner et al., 2016). Dickerson et al. (1995) measured CO concentrations of 0.114 ppm for marine sources and 0.157 ppm for continental sources arriving at Bermuda, but Dadashazar et al. (2021) reports lower median CO concentrations of 0.0886 ppm during the spring between 2015 and 2019. The median $CH_4$ (CO) concentration >900 km from LaRC across all non-golden flights is $1.951^{+0.023}_{-0.006}$ $(0.103^{+0.013}_{-0.005})$ ppm. While median $CH_4$ concentrations for all other flights are close to the golden flights value, CO decreased 21%, which is closer in value to report measurements at Bermuda (Dadashazar et al., 2021; Dickerson et al., 1995). The non-golden flights had less continental influence near Bermuda (Fig. 2, C-H), potentially lowering CO concentrations while $CH_4$ was less affected, possibly due to methane's longer residence time."

Comment #4 Pg 13 Lines 254

You state that you used 70 of 86 measurements of the smoke plume and found that the CH4/CO ratio is low enough to suggest biomass burning. How was that number of measurements chosen, was it due to their CO concentration being greater than 168 ppm? If that is the case, why was 168 ppm chosen as the cut-off? Some additional information could be useful explaining why those measurements were chosen.

Response: 168 ppm was chosen because there is a change in the slope $\Delta CH_4/\Delta CO$ as seen in Fig. 5 at that CO concentration. The slope of values below 168 ppm matched the slope of values measured outside the smoke layer. When CO concentrations went above 168 ppm, there was a visible change in $\Delta CH_4/\Delta CO$. 70 of the 86 measurements were above this threshold. We have updated the text for clarity:

"For values measured in the smoke layer (outlined in black in Fig. 5b), the slope $\Delta CH_4/\Delta CO$ decreases for CO concentrations greater than 0.168 ppm (blue line in Fig. 5b). Using the 70 CO measurements which are greater than 0.168 ppm out of the 86 total measurements made within the layer, $\Delta CH_4/\Delta CO$ decreases to 0.21±0.06, which is well below the 0.3 biomass burning threshold."

Comment #5 Pg 13 Lines 254-257

Related to the previous comment, do you have any comment on what the source of the potential biomass burning could be, especially since it seems to be a unique case in the flights reported in this study?

Response: We added Fig. S6 and the following text to the end of Section 3.3 to address this question:

"Figure S6 shows the HYSPLIT 3-day back-trajectories during the interception of the smoke layer, active fire points from NASA's Fire Information for Resource Management System (FIRMS), and smoke aerosol optical depth from the Navy Aerosol Analysis and Prediction System (NAAPS) (Lynch et al., 2016). FIRMS data shows that there were active fires in the central U.S. leading up to the flights, which fell along the HYSPLIT trajectories (Fig. S6, B). NAAPS data show enhancements in AOD attributed to smoke in the central U.S. about a day prior to the aircraft intersecting the smoke layer, which is coincident with the back-trajectories 24 hours before the intersection of the smoke layer (Fig. S6, C)."

[Figure]

**Figure S6: (a) HYSPLIT three-day back trajectories ending at the location of Falcon research flight 143 (22 March 2022) along its flight path while it intersected the smoke layer. Altitude of air mass trajectories shown with color bar and numbers represent the days prior to the initialized point of the back-trajectories. These trajectories are overlayed in panels b-c. (b) NASA's Fire Information for Resource Management System (FIRMS) active fire points in red from 20 May through 22 May 2024. (c) Speciated Navy Aerosol Analysis and Prediction System (NAAPS) aerosol optical depth (smoke in blue) at 18:00 UTC 21 March, approximately 1 day before the smoke layer was intercepted by the aircraft.**

Comment #6 Pg 14 Lines 271

Here you mention the particle number concentration leveling off roughly 900 km offshore. Are these numbers leveling off at the measurement of 1150 cm-3 and do you feel they are representative of general background in the region, or something related to the outflow that had been followed on the flight?

Response: We agree that this leveled-off value should be compared to background measurements in the same region. We added to the manuscript a comparison of this value to another measurement from ACTIVATE, which was made of a background marine air mass with the same instrument in the same region:

"This leveled-off particle concentration is still greater than marine background measurements near Bermuda of particles <100 nm made during ACTIVATE below 1 km, which were typically below 300 cm$^{-3}$ (Ajayi et al., 2024)."

Comment #7 Pg 23 Lines 407

I believe you mean "wet scavenging" instead of "wet savaging"

Response: Thank you for catching that. The typo has been corrected.

**Reviewer 2**

**Reviewer Summary**

This study examined the evolution of trace gas and aerosol properties off the U.S. East Coast on a basis of eight transit flights. The results are fruitful and are beneficial to our understanding of the air pollutants and greenhouse gases transition between continental and marine environments. I suggest that it can be published in ACP after minor revisions.

**Comments**

1. Regarding the new particle formation, the Figure S9 might be better than Figure 10 to discuss the NPF events. Much higher N3-10 nm close to continental areas may indicate the NPF events typically take place for continental air mass-dominated cases, but not marine.

Response: We agree that Fig. S9 is the better choice for discussing new particle formation, so we have put it in the manuscript and moved Fig. 10 to the SI document. Now the text reads:

"In terms of absolute number concentration between 3 and 10 nm, Fig. 10 shows a general offshore decline for all flights without much difference between the BL and FT. This may indicate a greater amount of new particle formation near the coast where higher concentrations of precursor gases exist compared to the marine environment."

2. R160 and R179 in Figure 13, the AMS results showed much higher sulfate mass fraction. RF 161 in Figure 14, ammonium mass fraction was dominated between 600-800 km. There is a lack of explanation.

Response: We agree these anomalies should be addressed in the manuscript. We added a discussion on the enhanced sulfate mass fraction for RFs 160 and 179:

"There is a significant increase in sulfate mass fraction for RFs 160 and 179 for distances from LaRC greater than 400 and 800 km, respectively. This corresponds to regions sampling marine aerosol as shown in the HYSPLIT modeling in Fig. 2. These marine air masses contained reduced concentrations of organics, thereby enhancing the sulfate mass fraction."

We also addressed the increase in ammonium mass fraction during RF 161:

"There is an increase in FT ammonium mass fraction and decrease in organic and sulfate mass fraction for RF 161 between 600 and 800 km corresponding to low total AMS mass concentration. HYSPLIT modeling shows the aircraft sampled an air mass originating over the midwestern U.S. unlike the rest of the flight, which mostly sampled air masses from the U.S. East Coast (Fig. 2, G). Agriculture emissions could be a possible explanation as they are a major source of ammonium aerosol in the midwestern U.S. (Stephen and Aneja, 2008)."

3. Typically, the particle hygroscopicity decreases with enhanced organic fraction. While, in Figure 17, some cases (e, f and i) showed a non-pronounced dependency between f(RH) and organic mass fraction. What are the reasons?

Response: Thank you for raising this question. We added a possible explanation to the manuscript: "Several flight days (Fig. 17; E, F & I) showed a less prominent relationship between organic mass fraction and f(RH). The nephelometers that are used to retrieve f(RH) can measure species that the AMS cannot, like black carbon and dust, which may have caused organic mass fraction to be less correlated with f(RH) for some flight days."

4. In Figure 19, there is spike of dnp/dlogDp below 10 nm. Is it an outlier?

Response: Yes, this is likely an outlier as the spike only appears in the 75th percentile of the measurements for that level leg. The median and 25th percentiles are zero in that same size bin. We did not feel like we needed to make a change to the manuscript for this comment.

**References:**

Ajayi, T. A., Choi, Y., Crosbie, E. C., DiGangi, J. P., Diskin, G. S., Fenn, M. A., Ferrare, R. A., Hair, J. W., Hilario, M. R. A., Hostetler, C. A., Kirschler, S., Moore, R. H., Shingler, T. J., Shook, M. A., Soloff, C., Thornhill, K. L., Voigt, C., Winstead, E. L., Ziemba, L., and Sorooshian, A.: Vertical variability of aerosol properties and trace gases over a remote marine region: A case study over Bermuda, EGUsphere, 2024, 1-35, https://doi.org/10.5194/egusphere-2024-1065, 2024.

Corral, A. F., Braun, R. A., Cairns, B., Gorooh, V. A., Liu, H., Ma, L., Mardi, A. H., Painemal, D., Stamnes, S., van Diedenhoven, B., Wang, H., Yang, Y., Zhang, B., and Sorooshian, A.: An Overview of Atmospheric Features Over the Western North Atlantic Ocean and North American East Coast – Part 1: Analysis of Aerosols, Gases, and Wet Deposition Chemistry, J Geophys Res-Atmos, 126, e2020JD032592, https://doi.org/10.1029/2020JD032592, 2021.

Dadashazar, H., Alipanah, M., Hilario, M. R. A., Crosbie, E., Kirschler, S., Liu, H., Moore, R. H., Peters, A. J., Scarino, A. J., Shook, M., Thornhill, K. L., Voigt, C., Wang, H., Winstead, E., Zhang, B., Ziemba, L., and Sorooshian, A.: Aerosol responses to precipitation along North American air trajectories arriving at Bermuda, Atmos. Chem. Phys., 21, 16121-16141, https://doi.org/10.5194/acp-21-16121-2021, 2021.

Dickerson, R. R., Doddridge, B. G., Kelley, P., and Rhoads, K. P.: Large-scale pollution of the atmosphere over the remote Atlantic Ocean: Evidence from Bermuda, J Geophys Res-Atmos, 100, 8945-8952, https://doi.org/10.1029/95JD00073, 1995.

Lynch, P., Reid, J. S., Westphal, D. L., Zhang, J., Hogan, T. F., Hyer, E. J., Curtis, C. A., Hegg, D. A., Shi, Y., Campbell, J. R., Rubin, J. I., Sessions, W. R., Turk, F. J., and Walker, A. L.: An 11-year global gridded aerosol optical thickness reanalysis (v1.0) for atmospheric and climate sciences, Geosci. Model Dev., 9, 1489-1522, 10.5194/gmd-9-1489-2016, 2016.

Stephen, K. and Aneja, V. P.: Trends in agricultural ammonia emissions and ammonium concentrations in precipitation over the Southeast and Midwest United States, Atmos Environ, 42, 3238-3252, https://doi.org/10.1016/j.atmosenv.2007.05.062, 2008.

Turner, A. J., Jacob, D. J., Benmergui, J., Wofsy, S. C., Maasakkers, J. D., Butz, A., Hasekamp, O., and Biraud, S. C.: A large increase in U.S. methane emissions over the past decade inferred from satellite data and surface observations, Geophys Res Lett, 43, 2218-2224, https://doi.org/10.1002/2016GL067987, 2016.